# The intralumenal fragment pathway mediates ESCRT-independent surface transporter down-regulation

Erin Kate McNally[1] & Christopher Leonard Brett[1]

Surface receptor and transporter protein down-regulation is assumed to be exclusively mediated by the canonical multivesicular body (MVB) pathway and ESCRTs (Endosomal Sorting Complexes Required for Transport). However, few surface proteins are known to require ESCRTs for down-regulation, and reports of ESCRT-independent degradation are emerging, suggesting that alternative pathways exist. Here, using *Saccharomyces cerevisiae* as a model, we show that the hexose transporter Hxt3 does not require ESCRTs for down-regulation conferring resistance to 2-deoxyglucose. This is consistent with GFP-tagged Hxt3 bypassing ESCRT-mediated entry into intralumenal vesicles at endosomes. Instead, Hxt3-GFP accumulates on vacuolar lysosome membranes and is sorted into an area that, upon fusion, is internalized as an intralumenal fragment (ILF) and degraded. Moreover, heat stress or cycloheximide trigger degradation of Hxt3-GFP and other surface transporter proteins (Itr1, Aqr1) by this ESCRT-independent process. How this ILF pathway compares to the MVB pathway and potentially contributes to physiology is discussed.

[1] Department of Biology, Concordia University, 7141 Sherbrooke St. W., SP-501.15, Montréal, QC H4R 1R6, Canada. Correspondence and requests for materials should be addressed to C.L.B. (email: christopher.brett@concordia.ca)

Surface polytopic proteins including receptors, transporters, and channels are internalized and sent to lysosomes for degradation[1–3]. Precise control of their surface levels underlies diverse physiology, including endocrine function, wound healing, tissue development, nutrient absorption, and synaptic plasticity[2,4–11]. Damaged surface proteins are also cleared by this mechanism to prevent proteotoxicity[12–14]. To trigger this process, surface proteins are labeled with ubiquitin—in response to changing substrate levels, heat stress to induce protein misfolding or cellular signaling for example—and then selectively internalized by the process of endocytosis[13,15–20]. Within the cell, they are sent to endosomes where they encounter ESCRTs (endosomal sorting complexes required for transport). These five protein complexes (ESCRT-0, -I, -II, -III, and the Vps4 complex) sort and package these internalized surface proteins into IntraLumenal Vesicles (ILVs)[3]. After many rounds, ILVs accumulate creating a mature multivesicular body (MVB)[21,22]. The MVB then fuses with lysosomes to expose protein laden ILVs to lumenal hydrolases for catabolism[2].

Although many examples of ESCRT-mediated protein degradation have been published[20], reports of ESCRT-independent degradation of surface proteins are emerging[23–27]. Furthermore, ILVs can be formed independent of ESCRT function and proteins recognized by ESCRTs continue to be degraded when ESCRTs are impaired[28–31]. These realizations have led to one of the most prominent open questions in our field: What accounts for ESCRT-independent ILV formation and surface protein degradation?

Around the time when ESCRTs were discovered[32], Wickner, Merz and colleagues reported that an ILV-like structure called an intralumenal fragment (ILF) is formed as a byproduct of homotypic vacuolar lysosome (or vacuole) fusion in the model organism *Saccharomyces cerevisiae*[33]. Prior to lipid bilayer merger, fusogenic proteins and lipids concentrate within a ring at the vertex between apposing vacuole membranes[34,35]. Upon SNARE (Soluble NSF [N-ethylmaleimide-sensitive factor] Associated protein REceptor)-mediated membrane fusion at the vertex ring, the encircled area of membrane, called the boundary, is excised and internalized within the lumen of the fusion product where it encounters acid hydrolases[36].

We recently discovered that vacuolar polytopic proteins, e.g., ion and nutrient transporters, are selectively sorted into the boundary membrane for degradation in response to substrate levels, misfolding by heat stress or TOR (Target Of Rapamycin) signaling triggered by cycloheximide[37]. Named the ILF pathway, this process functions independently of ESCRTs and instead relies on the fusion protein machinery for transporter sorting and ILF formation. This process performs similar functions as ESCRTs, except the mechanisms underlying protein sorting and packaging are distinct. Thus, the possibility exists that surface polytopic proteins may also be degraded by the ILF pathway if they can be delivered to the vacuole membrane after internalization.

Can internalized surface proteins be delivered to vacuole membranes? To our knowledge, this hypothesis has not been formally tested. However, this proposition seems reasonable when considering the consequence of bypassing ESCRT function in the canonical MVB pathway (Fig. 1a). In theory, any internalized surface polytopic protein that is not packaged into ILVs by ESCRTs (or returned to the surface) remains embedded within the outer membrane of the mature MVB. When this membrane merges with the vacuole membrane upon heterotypic fusion, these proteins are then exposed to the ILF machinery, which may package them for degradation. Thus, by simply avoiding recognition by ESCRTs, internalized surface proteins are, by default, delivered to vacuole membranes and the ILF pathway.

Upon reexamination of micrographs presented in earlier reports on receptor and transporter down-regulation, we found that some internalized surface polytopic proteins appear on vacuole membranes on route to the lumen for degradation, e.g., the high-affinity tryptophan permease Tat2[38], peptide transporter Ptr2[39], myo-inositol transporter Itr1[40], and glucose transporters Hxt1 and Hxt3[41]. We also noticed that most of these published studies do not directly assess whether ESCRTs are required for protein degradation. Given that internalized surface transporters and receptors can appear on vacuole membranes, we decided to test the hypothesis that the ILF pathway represents an alternative, ESCRT-independent mechanism for degradation of surface polytopic proteins (Fig. 1a).

## Results

### ESCRTs are not required for 2-deoxyglucose resistance.
ESCRTs have been implicated in the down-regulation of surface transporters required for *S. cerevisiae* cell survival and proliferation in the presence of toxic substrates. For example, to prevent entry of the toxic arginine analog canavanine, the surface arginine permease Can1 is endocytosed and sorted for degradation by ESCRTs[17]. Similarly, the surface glucose transporter Hxt3 is internalized and degraded in the presence of 2-deoxyglucose, a toxic glucose analog[41]. It has been proposed that deleting ESCRTs blocks delivery to vacuoles and subsequent degradation of these transporters, causing them to accumulate in aberrant endosomal structures. Here they can be returned to the plasma membrane by a Snx3-dependent retrograde trafficking pathway, allowing toxin entry[17]. Thus, based on this model, deletion of ESCRT genes should reduce cell viability in the presence of canavanine or 2-deoxyglucose.

We tested this hypothesis by treating yeast cultures with increasing concentrations of either toxin and then assessed effects on cell viability by imaging and counting dead yeast cells stained with methylene blue (Fig. 1b). As expected, deleting components of ESCRT-0 (*VPS27*) or ESCRT-II (*VPS36*)—to disrupt different early- or late-acting complexes, respectively, necessary for protein sorting into ILVs—enhanced sensitivity to the toxin canavanine. However, deleting these ESCRT genes had no effect on cell viability in the presence of 2-deoxyglucose. As controls, we examined cells lacking *CAN1* or *HXT3* and found that they were resistant to canavanine or 2-deoxyglucose, respectively, as predicted. Thus, these data suggest that ESCRTs mediate degradation of Can1 but may not be required for Hxt3 degradation triggered by 2-deoxyglucose.

### Hxt3 protein degradation is ESCRT-independent.
Based on micrographs presented in previous reports showing Hxt3 on vacuole membranes[41], we first assessed the possibility that internalized Hxt3 bypassed ESCRTs altogether at the endosome and instead were delivered to vacuole membranes where it may be sorted for degradation. To test this hypothesis, we used fluorescence microscopy to monitor the distribution of GFP-tagged Hxt3 in live *S. cerevisiae* cells over time after addition of 2-deoxyglucose (Fig. 1c). As predicted, Hxt3-GFP is exclusively found on the plasma membrane before treatment. After addition of 2-deoxyglucose, it first appears on puncta (representing endosomes; at 5 min) and later on vacuole membranes stained with FM4–64 (30 min; Fig. 1d). Finally, Hxt3-GFP accumulates within the vacuole 60–120 min after treatment. We assessed all cells imaged, quantified these observations (Fig. 1e) and confirmed that Hxt3-GFP was progressively cleared from the plasma membrane while accumulating in puncta, on the vacuole membrane and in the vacuole lumen over time after 2-deoxyglucose addition. In contrast,

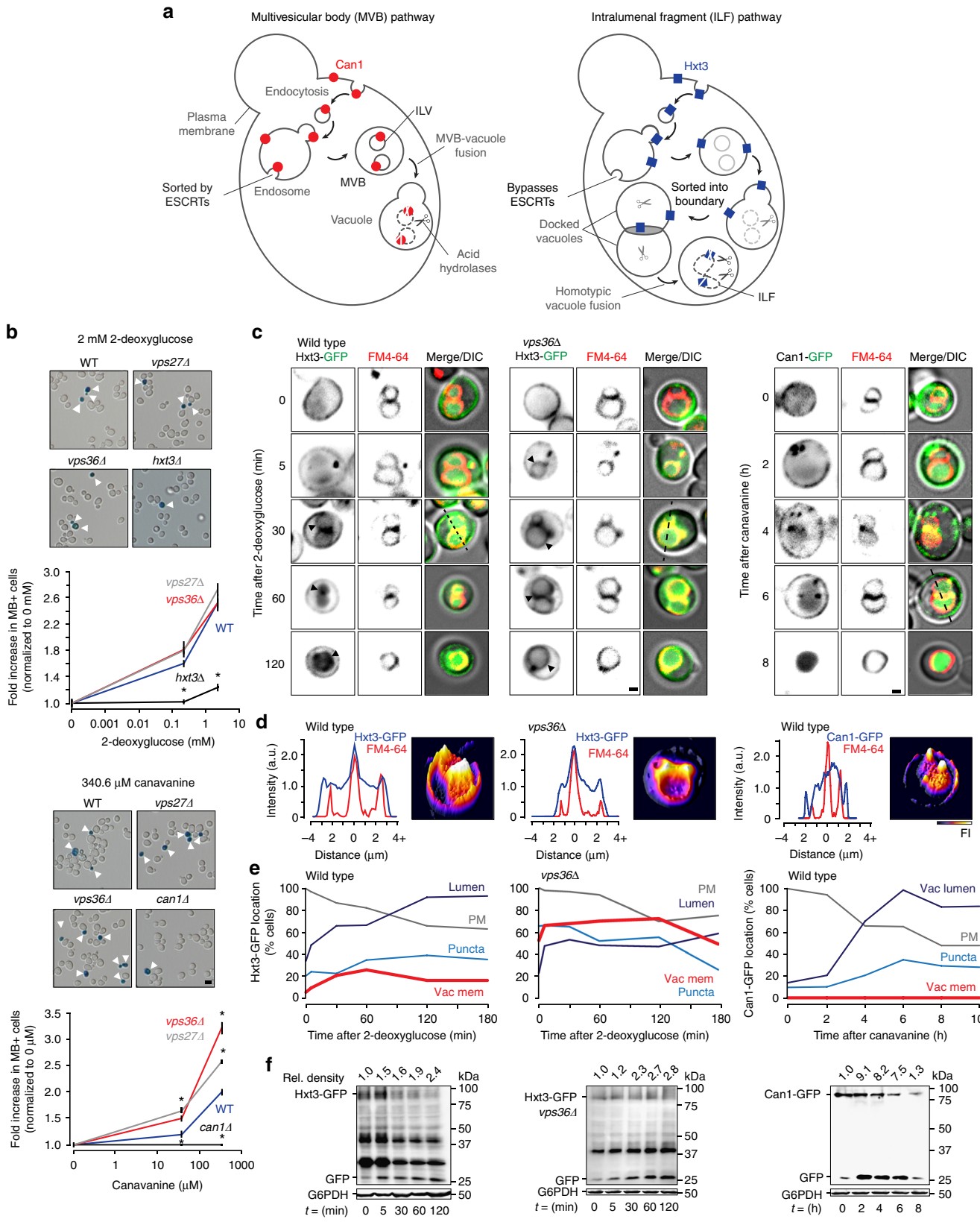

internalized Can1-GFP, an ESCRT-client, never appears on vacuole membranes on route to the vacuole lumen for degradation when cells were treated with canavanine (Fig. 1c–e), as predicted for the canonical MVB pathway. To confirm that proteolysis occurs, we conducted western blot analysis to detect

cleavage of GFP from Hxt3 or Can1 in whole-cell lysates (Fig. 1f). As predicted, we found that more GFP was cleaved over time correlating with toxin triggered downregulation of these transporters regardless of route taken to the vacuole lumen.

**Fig. 1** Internalized transporter proteins take two routes to the vacuole lumen for degradation. **a** Cartoon illustrating how surface transporter proteins Can1 or Hxt3 are sorted for degradation by MVB or ILF pathways, respectively. **b** Micrographs of yeast treated with the toxin 2-deoxyglucose (top) or canavanine (bottom) and stained with methylene blue to detect dead (MB+) cells indicated by white arrowheads. Graphs show relative number of dead cells observed at increasing concentrations of either toxin. Means ± S.E.M. plotted. *P < 0.05, as compared to WT values by t-test. For 2-deoxyglucose treatment, n = 3 experiments, whereby a total of 2355 WT, 2105 hxt3Δ, 1,786 vps27Δ and 1413 vps36Δ cells were analyzed. For canavanine treatment, n = 3 experiments whereby 2671 WT, 2033 can1Δ, 1647 vps27Δ and 1,538 vps36Δ cells were analyzed. **c** Micrographs showing route taken by Can1-GFP from the surface to the vacuole lumen in response to 340.6 μM canavanine over 8 h in live wild type cells (right) or Hxt3-GFP in response to 0.2 mM 2-deoxyglucose over 120 min in live wild type (left) or vps36Δ (middle) cells. All cells were treated with FM4–64 to label vacuole membranes. Black arrowheads indicate GFP on the vacuole membrane. **d** Three-dimensional GFP fluorescence intensity (FI) plots and line plots of Can1-GFP (right), Hxt3-GFP (middle, left) or FM4–64 fluorescence intensity for micrographs with lines shown in **c**, to indicate vacuole membrane localization. Can1-GFP vacuole distribution is shown at 6 h because intensity was too low at 4 h to generate informative plots. **e** Using micrographs shown in **c**, we calculated the proportion of wild type or vps36Δ cells showing Can-GFP (right) or Hxt3-GFP (middle, left) fluorescence on the plasma membrane (PM), intracellular puncta, vacuole membrane (Vac mem) or vacuole lumen over time after treatment with canavanine or 2-deoxyglucose. For Hxt3-GFP at 0, 5, 30, 60, 120 min, n = 195, 198, 200, 207, 214 WT cells, and n = 202, 205, 207, 266, 211 vps36Δ cells analyzed. For Can1-GFP at 0, 2, 4, 6, 8 h, n = 207, 210, 246, 220, 208 WT cells analyzed. **f** Western blot analysis of whole-cell lysates prepared from wild type (left) or vps36Δ (middle) cells expressing Hxt3-GFP before (0 min) or 5–120 min after treatment with 0.2 mM 2-deoxyglucose or wild-type cells expressing Can1-GFP before or after treatment with 340.6 μM canavanine. Blots were stained for GFP or glucose-6-phosphate dehydrogenase (G6PDH; as load controls). Estimated molecular weights and cleaved GFP band densities relative to 0 min normalized to load controls are shown. Blots shown are representatives of n = 5 experiments. Scale bars, 1 μm

In all, these findings suggest that internalized Hxt3-GFP bypasses ESCRT function at endosomes, and instead is delivered to vacuole membranes on route to the lumen where it is degraded in response to 2-deoxyglucose. If true, then deleting ESCRT genes should have no effect on this process. To test this hypothesis, we tracked the subcellular distribution of Hxt3-GFP after 2-deoxyglucose treatment in cells lacking the ESCRT gene VPS36 (Fig. 1c–e). Although Hxt3-GFP continues to be delivered to the vacuole lumen over time, we noticed that on route, Hxt3-GFP abnormally accumulated on puncta and vacuole membranes, even prior to treatment, in mutant cells. This is consistent with previous reports showing that deleting ESCRT genes impairs, but does not entirely block, endocytosis and delivery of vacuole cargo proteins in addition to completely abolishing protein sorting into ILVs[3]. Importantly, GFP continued to be cleaved from Hxt3 (Fig. 1f) confirming that it continues to be degraded in absence of VPS36. Thus, delivery of internalized Hxt3-GFP to the vacuole lumen for degradation in response to 2-deoxyglucose does not require ESCRTs.

**2-deoxyglucose triggers Hxt3 degradation by ILF pathway.** What mediates ESCRT-independent Hxt3-GFP degradation? Because this surface polytopic proteins appear on the vacuole membrane, we reasoned that this is where it is sorted and packaged into intralumenal vesicles or fragments for degradation. Although microautophagy may be responsible, no intermediate structures resembling elongated invaginations extending into the vacuole lumen were observed[42] and there are no reports of selective protein sorting associated with this process (rather these structures seem to be devoid of membrane proteins[42]). Instead, we considered two processes reported to mediate selective degradation of vacuole membrane proteins: the ESCRT-dependent vReD (vacuole membrane Recycling and Degradation) pathway[43] and ESCRT-independent ILF pathway[37]. Given that ESCRTs were not required for Hxt3-GFP down-regulation, we eliminated the possibility that the vReD pathway was responsible and focused on the ILF pathway.

Upon further examination of micrographs showing Hxt3-GFP distributed on vacuole membranes after toxin addition, we found that it was present within boundary membranes between docked vacuoles, the area of membrane that is internalized as an ILF and degraded upon fusion (Fig. 2a, b). To confirm, we measured the GFP fluorescence within the vacuole boundary membranes in the cell population (Fig. 2c). To determine if Hxt3-GFP was getting sorted into the boundary, we compared its distribution to three

GFP-tagged ILF-client proteins that are residents of the vacuole membrane: Fet5 (a copper oxidase) which is known to be excluded from boundaries, Vph1 (the stalk domain of the V-type H+-ATPase) which is distributed uniformly on vacuole membranes, and Fth1 (an iron transporter) which is enriched in boundaries (Fig. 2b–d; see ref. [37]). From this analysis, it was clear that the Hxt3-GFP boundary profile was most similar to Fth1-GFP (Fig. 2b, c), suggesting that it was enriched and thus possibly sorted into the ILF pathway.

We also found that the distribution of these three GFP-tagged resident vacuole proteins was unaffected by 2-deoxyglucose (Fig. 2d), suggesting that their turnover was not stimulated by this toxin. Similarly, 2-deoxyglucose does not trigger down-regulation of GFP-tagged Can1 or the surface myo-inositol transporter Itr1 (Fig. 2d). Moreover, Hxt3-GFP was not down-regulated when cells were treated with canavanine (Fig. 2a). We assessed all cells imaged, quantified these observations and confirmed that only Hxt3-GFP was downregulated by 2-deoxyglucose but it was unaffected by canavanine (Fig. 2e). These important findings imply that Hxt3-GFP was selectively sorted into the ILF pathway in response to only its cognate toxic substrate 2-deoxyglucose.

Finally, to demonstrate that Hxt3-GFP was internalized into the lumen during fusion we recorded homotypic vacuole membrane fusion events in live cells treated with 2-deoxyglucose, and confirmed that Hxt3-GFP decorated ILFs created by fusion events (Fig. 2f; Supplementary Movie 1). Homotypic vacuole fusion was not stimulated by 2-deoxyglucose, as we counted the same number of fusion events within cells in the absence (CTL) and presence of the toxin (Fig. 2g). Thus, vacuole fusion persists under these conditions and seems to mediate the selective degradation of Hxt3 after endocytosis.

Together, these observations led us to conclude that Hxt3 avoids ESCRTs and is instead selectively sorted for degradation by the ILF pathway. Two questions immediately arose after making this exciting discovery: Is surface protein degradation by the ILF pathway only triggered by toxic substrates, or does it contribute to other triggers of surface transporter down-regulation? Is the ILF pathway responsible for degradation of other surface polytopic proteins? In other words, how pervasive is the contribution of the ESCRT-independent ILF pathway to surface transporter and receptor protein downregulation?

**ILF pathway degrades Hxt3 in response to diverse stimuli.** When yeast cells are challenged with acute heat stress, surface

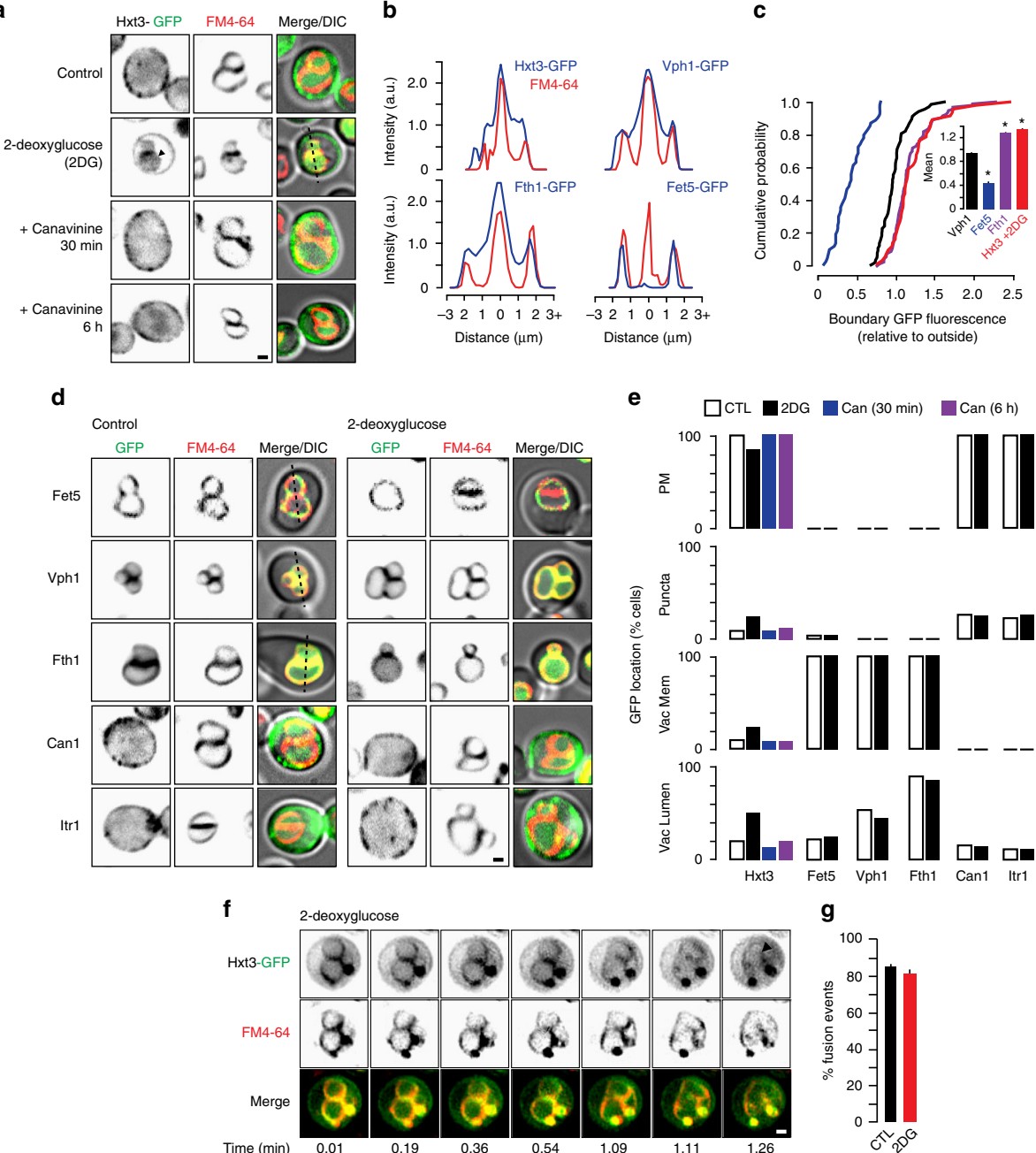

**Fig. 2** Hxt3-GFP is selectively degraded by the ILF pathway in response to 2-deoxyglucose. **a** Micrographs of live wild type cells expressing GFP-tagged Hxt3 after treatment with 2-deoyglucose for 30 min, or canavanine for 30 min or 6 h. Vacuole membranes were stained with FM4–64. Arrowhead indicates GFP on the vacuole membrane. **b** Line plots of GFP or FM4–64 fluorescence intensity for lines shown in **a** and **d**, to indicate vacuole membrane localization. GFP values greater than the FM4–64 signal at boundaries (near 2, for two membranes) indicate enrichment. **c** Using micrographic data presented in **a** and **d**, we generated cumulative probability plots of GFP- tagged Hxt3, Fet5, Vph1 or Fth1 fluorescence measured within the boundary membrane of docked vacuoles within live wild-type cells in the absence or presence of 2-deoxyglucose (2DG). Averages ± S.E.M. are shown in insets. *$P >$ 0.05, as compared to Vph1 by $t$-test. $n = 3$ experiments whereby 72 Vph1-GFP, 77 Fet5-GFP, 107 Fth1-GFP or 82 Hxt3-GFP + 2DG boundaries within cells were analyzed. **d** Micrographs of live wild type cells expressing GFP-tagged resident vacuole transporters Fet5, Vph1 or Fth1, or surface transporters Can1 or Itr1, before (control) after treatment with 2-deoxyglucose for 30 min. Vacuole membranes were stained with FM4–64. **e** Using micrographs shown in **a** and **d**, we calculated the proportion of wild type cells showing GFP fluorescence on the plasma membrane (PM), intracellular puncta, vacuole membrane (Vac Mem) or vacuole lumen (Vac Lumen) after treatment with 2-deoxyglucose or canavanine. For control conditions, $n = 218$ Hxt3-GFP, 187 Fet5-GFP, 265 Vph1-GFP, 267 Fth1-GFP, 233 Can1-GFP, and 189 Itr1-GFP cells. After 2-deoxyglucose treatment, $n = 200$ Hxt3-GFP, 143 Fet5-GFP, 128 Vph1-GFP, 161 Fth1-GFP, 150 Can1-GFP, and 178 Itr1-GFP cells. For canavanine treatment, $n = 171$ or 166 Hxt3-GFP cells after 0.5 or 6 h, respectively. **f** Images from time-lapse video showing a homotypic vacuole fusion event within a live wild-type cell expressing Hxt3-GFP stained with FM4–64 to label vacuole membranes and treated with 2-deoxyglucose. Arrowhead indicates newly formed ILF. See Supplementary Movie 1. **g** Analysis of micrographic data shown in **f** showing the proportion of cells that displayed a vacuole fusion event within 5 min in the absence (CTL) or presence of 2-deoxyglucose (2DG). Averages ± S.E.M. are shown. $P > 0.05$, when 2DG was compared to CTL by $t$-test. $n = 5$ experiments whereby a total of 121 (CTL) or 174 (2DG) Hxt3-GFP cells were analyzed. Scale bars, 1 μm

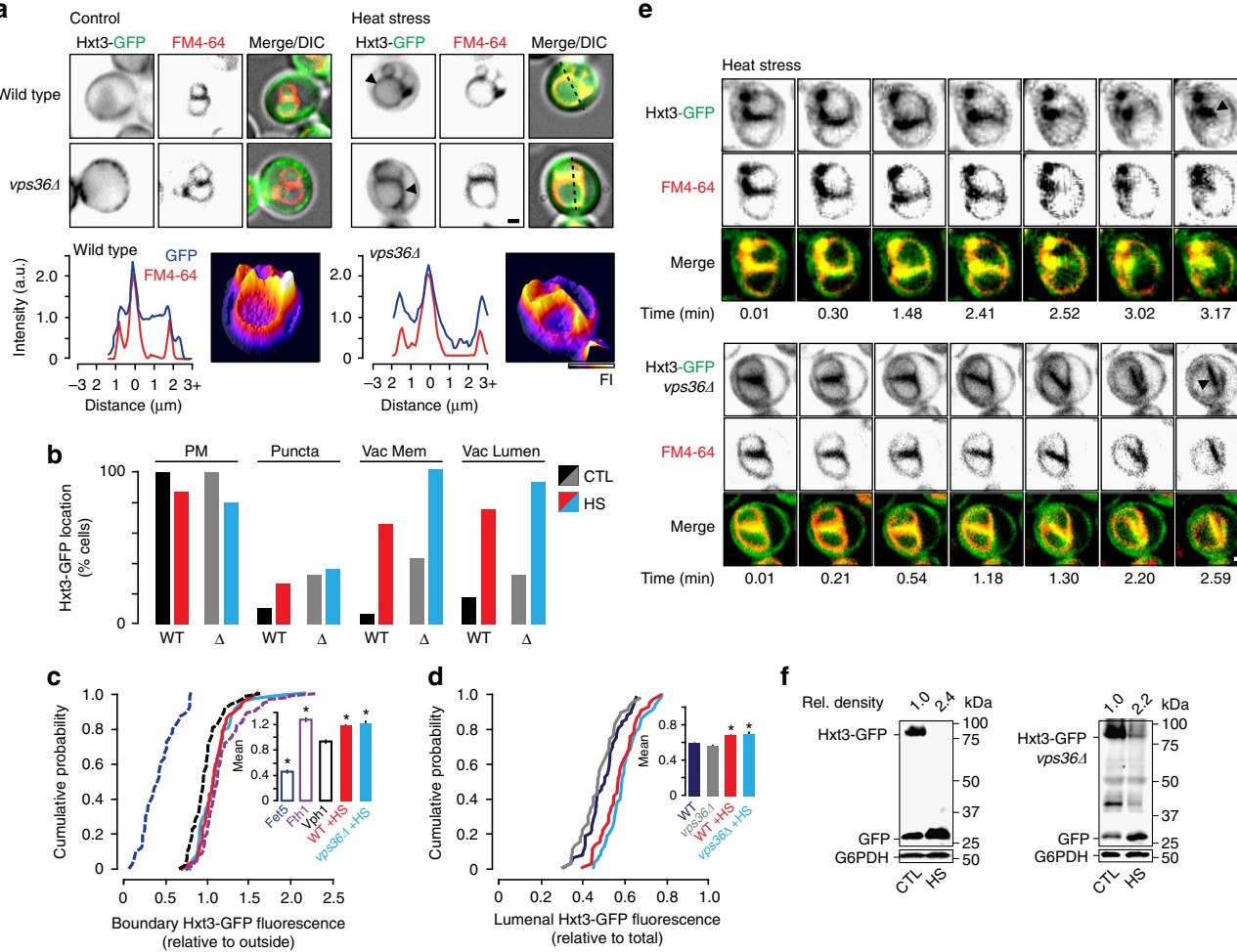

**Fig. 3** Quality control of Hxt3-GFP is mediated by the ILF pathway. **a** Fluorescence and DIC micrographs of live wild type or *vps36Δ* cells expressing GFP-tagged Hxt3 before (control) or after heat stress (37 °C for 15 min). Vacuole membranes were stained with FM4–64. Arrowheads indicate GFP on the vacuole membrane. 3-dimensional GFP fluorescence intensity (FI) plots and line plots of Hxt3-GFP or FM4–64 fluorescence intensity for lines shown in **a**, to indicate vacuole membrane localization after heat stress. **b** Micrographic data shown in **a** was used to calculate the proportion of wild type or *vps36Δ* cells that show Hxt3-GFP fluorescence on the plasma membrane (PM), intracellular puncta, vacuole membrane (Vac Mem) or vacuole lumen (Vac Lumen) before or after treatment with heat stress. Under control (CTL) conditions, *n* = 218 WT cells or 243 *vps36Δ* cells under control conditions (CTL); *n* = 163 WT cells or 196 *vps36Δ* cells after heat stress. **c, d** Micrographic data shown in **a** was used to generate cumulative probability plots of Hxt3-GFP fluorescence measured within the boundary membrane (**c**) or lumen (**d**) of vacuoles within live wild type or *vps36Δ* cells before or after heat stress (HS). GFP-tagged Fet5 (excluded), Vph1 (ubiquitous) and Fth1 (enriched) are shown for reference (see Fig. 2D). Averages ± S.E.M. are shown in insets. *P < 0.05, as compared to Vph1 by *t*-test. *n* = 3 experiments whereby a total of 81 WT or 102 *vps36Δ* cells were analyzed under control conditions, and 75 WT or 88 *vps36Δ* cells were analyzed after heat stress. **e** Images from time-lapse videos showing homotypic vacuole fusion events within live wild type or *vps36Δ* cell expressing Hxt3-GFP treated with heat stress. Vacuole membranes were stained with FM4–64. Arrowheads indicate newly formed ILFs. See Supplementary Movies 2 and 4. Example shown is a representation of *n* = 4 experiments. **f** Western blot analysis of whole-cell lysates prepared from wild-type or *vps36Δ* cells expressing Hxt3-GFP after heat stress (HS) stained with anti-GFP antibody. Estimated molecular weights and cleaved GFP band densities relative to CTL normalized to load controls (G6PDH) are shown. Blots shown are representatives of *n* = 3 experiments. Scale bars, 1 μm

polytopic proteins are thought to misfold, which disrupts their function and contributes to membrane permeabilization preceding cell death. If they cannot be refolded (through the activity of chaperones), they are internalized and sent to the vacuole lumen for degradation—presumably by the MVB pathway—to mediate surface polytopic protein quality control[20].

Thus, to answer the first question, we tested whether treating cells with heat stress also triggers Hxt3-GFP degradation by the ILF pathway using fluorescence microscopy to monitor its distribution within living cells. As with 2-deoxyglucose, we found that Hxt3-GFP is internalized from the surface after heat stress and accumulates on vacuole membranes (Fig. 3a, b) within boundaries between docked organelles (Fig. 3a, c) and in the vacuole lumen (Fig. 3d). We confirmed that Hxt3-GFP decorated ILFs formed

during homotypic vacuole fusion events in live cells (Fig. 3e; Supplementary Movie 2), which correlated with enhanced Hxt3-GFP degradation as assessed by western blot analysis of whole-cell lysates (Fig. 3f). Similar observations were made when we applied cycloheximide (Fig. 4; Supplementary Movie 3), a trigger of TOR (Target Of Rapamycin) signaling, an important mediator of cell growth and metabolism[44], that stimulates surface protein degradation[17] and the ILF pathway[37]. Deleting *VPS36* had no effect on Hxt3-GFP clearance from plasma membranes, delivery to vacuole boundary membranes, internalization during homotypic vacuole fusion or protein degradation in response to heat stress (Fig. 3; Supplementary Movie 4) or cycloheximide (Fig. 4; Supplementary Movie 5), confirming that this process is ESCRT-independent. Thus, we concluded that the ILF pathway mediates Hxt3 protein

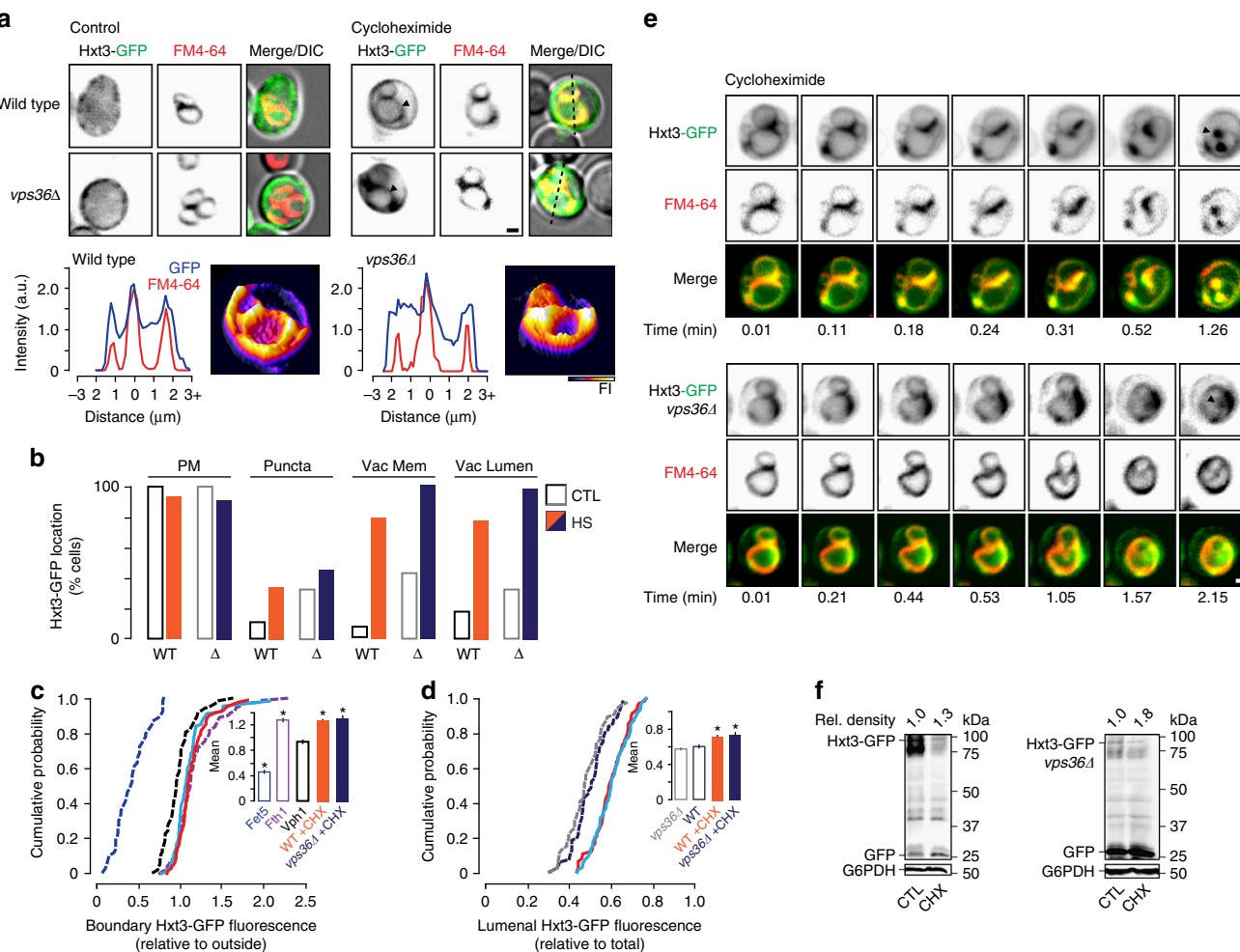

**Fig. 4** Cycloheximide triggers Hxt3-GFP degradation by the ILF pathway. **a** Fluorescence and DIC micrographs of live wild-type or *vps36Δ* cells expressing GFP-tagged Hxt3 before (control) and after treatment with 100 μM cycloheximide for 45 min. Vacuole membranes were stained with FM4-64. Arrowheads indicate GFP on the vacuole membrane. Three-dimensional GFP fluorescence intensity (FI) plots and line plots of Hxt3-GFP or FM4-64 fluorescence intensity for lines shown in **a**, to indicate vacuole membrane localization after cycloheximide treatment. **b** Micrographic data shown in **a** was used to calculate the proportion of wild type (*n* = 142) or *vps36Δ* (*n* = 158) cells that show Hxt3-GFP fluorescence on the plasma membrane (PM), intracellular puncta, vacuole membrane (Vac Mem) or vacuole lumen (Vac Lumen) after treatment with cycloheximide. Values before cycloheximide treatment are shown for reference (see Fig. 3b). **c, d** Micrographic data shown in **a** was used to generate cumulative probability plots of Hxt3-GFP fluorescence measured within the boundary membrane (**c**) or lumen (**d**) of vacuoles within live wild-type or *vps36Δ* cells after treatment with cycloheximide (CHX). GFP-tagged Fet5 (excluded), Vph1 (ubiquitous), and Fth1 (enriched; see Fig. 2d) and lumenal values before cycloheximide treatment (see Fig. 3d) are shown for reference. Averages ± S.E.M. are shown in insets. *P < 0.05, as compared to Vph1 by *t*-test. *n* = 5 experiments whereby a total of 73 wild type cells or 108 *vps36Δ* cells after CHX treatment were analyzed. **e** Images from time-lapse videos showing homotypic vacuole fusion events within live wild type or *vps36Δ* cell expressing Hxt3-GFP treated with cycloheximide. Vacuole membranes were stained with FM4-64. Arrowheads indicate initial fusion sites leading to ILF formation. See Supplementary Movies 3 and 5. Examples shown are representatives of *n* = 4 experiments. **f** Western blot analysis of whole-cell lysates prepared from wild type or *vps36Δ* cells expressing Hxt3-GFP before (CTL) or after treatment with cycloheximide (CHX) stained with anti-GFP antibody. Estimated molecular weights and cleaved GFP band densities relative to CTL normalized to load controls (G6PDH) are shown. Blots shown are representatives of *n* = 4 experiments. Scale bars, 1 μm

quality control and degrades this transporter protein in response to cycloheximide.

**ILF pathway machinery mediates Hxt3-GFP degradation in vitro.** To verify that the observed degradation of internalized Hxt3-GFP was conducted by the ILF pathway machinery, we repeated experiments using in vitro fusion reactions containing isolated vacuoles based on the following reasoning [see [37]]: (1) all molecular machinery underlying the ILF pathway copurifies with vacuoles, in a preparation that excludes possible cytoplasmic contributors to Hxt3-GFP degradation (e.g., the proteasome) as well as the protein translation machinery to eliminate interference

by biosynthesis. (2) activation of the Rab GTPase Ypt7 is critical for sorting proteins into the ILF pathway. However, deleting *YPT7* chronically blocks all vacuole fusion events including MVB-vacuole fusion preventing study of the ILF and MVB pathways in live cells. Instead, purified protein inhibitors of Ypt7–rGdi1 (a Rab-GTPase chaperone protein that extracts Ypt7 from membranes) and rGyp1-46 (the catalytic domain of the Rab GTPase activating protein Gyp1 that inactivates Ypt7[45]) can be added to in vitro vacuole fusion reactions containing healthy organelles to acutely block Ypt7 permitting study of protein sorting into ILFs. Thus, if the ILF machinery is responsible for Hxt3-GFP degradation, then it should continue to occur in vitro and be sensitive to Ypt7 inhibitors.

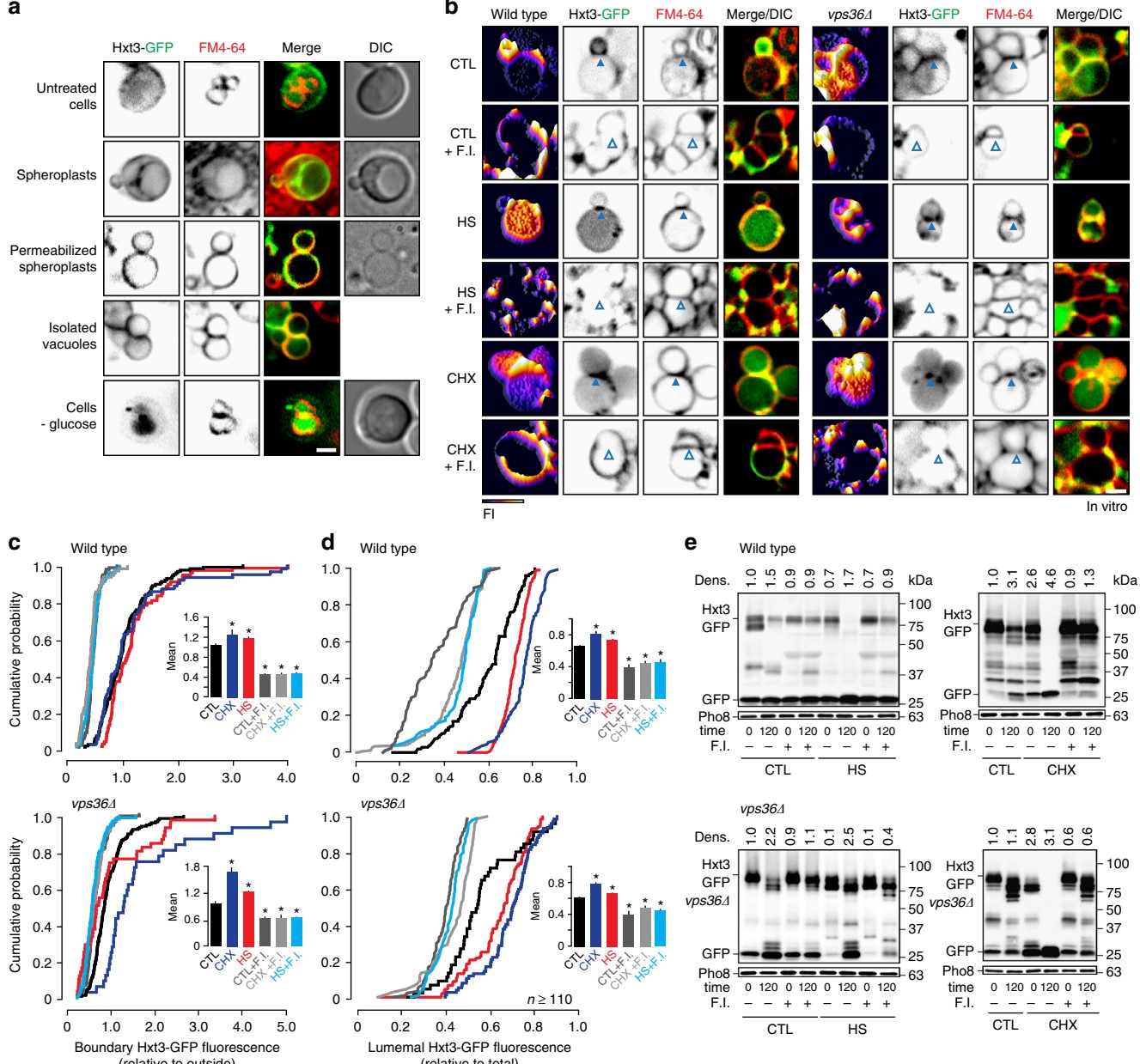

**Fig. 5** The ILF pathway machinery is responsible for Hxt3-GFP degradation. **a** Fluorescence and DIC micrographs of live wild-type cells expressing GFP-tagged Hxt3 before (untreated cells) and at stages of the vacuole isolation procedure, including after oxalyticase treatment to form spherophasts, after DEAE-dextran treatment to permeabilize spheroplasts, and isolated vacuoles. Untreated cells in SC medium were also withdrawn from glucose for 5 min. Vacuole membranes were stained with FM4-64. Micrographs shown are examples from $n = 3$ experiments. **b** Fluorescence micrographs of vacuoles isolated from wild type (left) or *vps36Δ* (right) cells after 30 min of fusion in the absence (CTL) or presence of heat stress (HS) or cycloheximide (CHX) with or without fusion inhibitors (4 μM rGdi1 and 3.2 μM rGyp1–46; F.I.). Vacuole membranes were stained with FM4–64 and 3-dimensional fluorescence intensity (FI) plots of Hxt3-GFP are shown. Boundary membranes indicating Hxt3-GFP enrichment (closed arrowheads) and exclusion (open arrowheads) are shown. **c**, **d** Micrographic data shown in **b** was used to generate cumulative probability plots of Hxt3-GFP fluorescence measured within the boundary membrane (**c**) or lumen (**d**) of vacuoles isolated from wild-type (top) or *vps36Δ* (bottom) cells before (CTL) and after treatment with HS or CHX with or without fusion inhibitors (F.I.). Means ± S.E.M. are shown in insets. *$P < 0.05$, as compared to CTL by *t*-test. $n = 3$ experiments whereby a total of 113, 108, 120, 110, 108, 107 WT vacuoles or 133, 110, 128, 112, 116, 109 *vps36Δ* vacuoles were analyzed under CTL, CTL + FI, HS, HS + F.I., CHX, CHX + F.I. conditions, respectively. **e** Western blot analysis of Hxt3-GFP degradation before (0 min) or after (120 min) vacuoles isolated from wild type (top) or *vps36Δ* (bottom) cells underwent fusion in the absence (CTL) or presence of HS (left) or CHX (right) with or without pretreatment with fusion inhibitors (F.I.). Estimated molecular weights and cleaved GFP band densities (Dens.) relative to 0 min normalized to load controls (Pho8) are shown. Representatives of $n = 4$ experiments are shown. Scale bars, 2 μm

To test this hypothesis, we first isolated vacuoles from untreated wild-type cells, imaged them and found that Hxt3-GFP decorated their membranes (Fig. 5a). Further investigation revealed that surface Hxt3-GFP is down-regulated in live yeast cells during the organelle isolation procedure (e.g., during spheroplasting when their cell walls are enzymatically removed; Fig. 5a). We made a similar observation when live cells were withdrawn from glucose for 5 min in growth medium (Fig. 5a), a

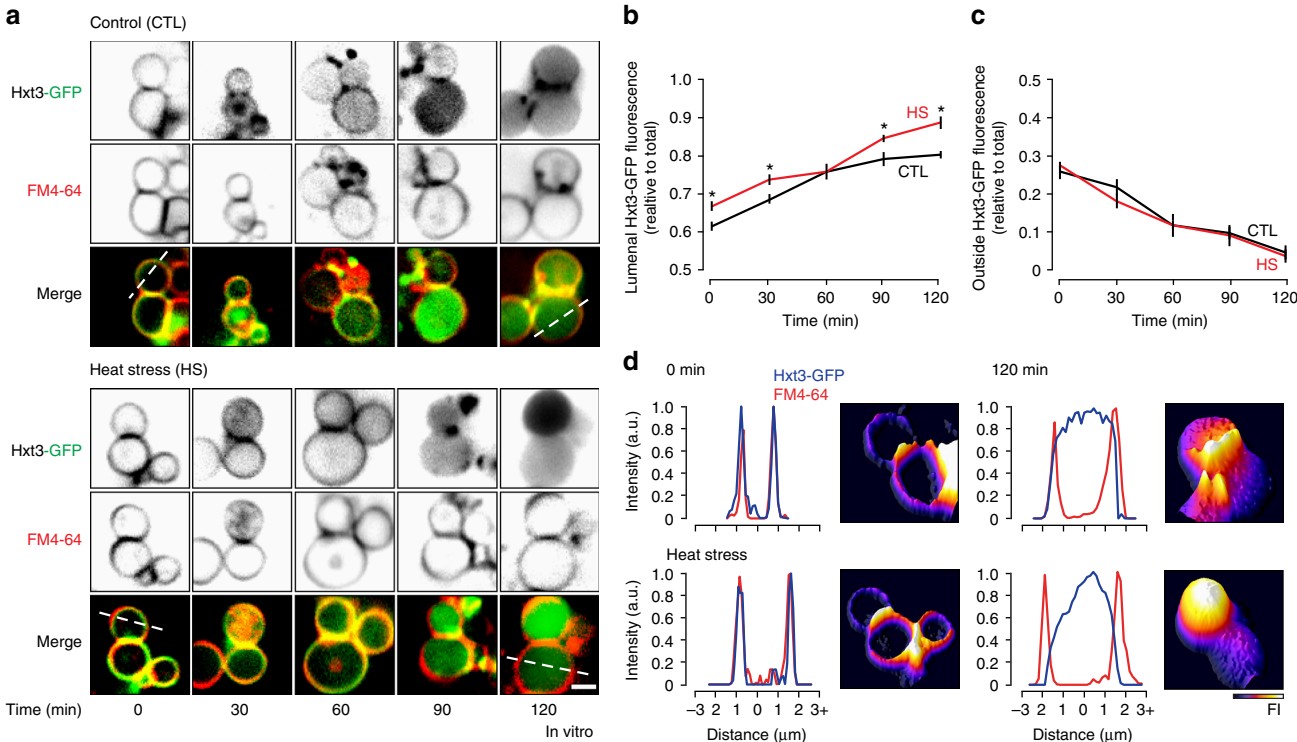

**Fig. 6** Hxt3-GFP is cleared from vacuole membranes during fusion in vitro. **a** Fluorescence micrographs of vacuoles isolated from wild type cells expressing Hxt3-GFP acquired over the course of the in vitro fusion reaction in the absence (control; left) or presence (right) of heat stress. Vacuole membranes were stained with FM4–64. Scale bar, 2 μm. **b**, **c** Using micrographic data shown in **a**, averages ± S.E.M. of Hxt3-GFP fluorescence intensity were measured within the lumen (**b**) or outside membrane (**c**) of vacuoles isolated from wild-type cells over the course of the fusion reaction with or without heat stress. *$P < 0.05$, as compared to CTL by $t$-test. $n = 5$ experiments whereby a total of 89, 88, 118, 100, 85 vacuoles were analyzed under CTL conditions, and 85, 108, 116, 110, 108 vacuoles were analyzed after HS, at 0, 30, 60, 90, 120 min, respectively. **d** Line plots of Hxt3-GFP and FM4-64 fluorescence, and 3-dimensional GFP fluorescence intensity (FI) plots of from micrographs in **a** showing vacuoles isolated from wild type cells before (0 min) and after 120 min of fusion after heat stress, to indicate that Hxt3-GFP is completely cleared from vacuole membranes over time in vitro

novel trigger of Hxt3-GFP degradation. All solutions for the organelle isolation procedure are devoid of glucose. Thus, we reasoned that glucose depletion during vacuole purification triggers Hxt3-GFP downregulation, accounting for its presence on vacuole membranes and within the lumen in vitro.

Next, we added ATP to isolated vacuoles to initiate homotypic fusion in vitro and imaged reactions 30 min afterwards (Fig. 5b). As expected, we found that Hxt3-GFP was present in boundary membranes (Fig. 5c) and within the vacuole lumen (Fig. 5d). Consistent with these findings, we found that more Hxt3-GFP was degraded after fusion by western blot analysis (Fig. 5e). However, we noted that relative high levels of cleaved GFP were found in all preparations containing isolated vacuoles, including those analyzed prior to fusion (0 min). It is possible that vacuole rupture and release of lumenal proteases may account for this observation. Although we cannot completely eliminate this possibility, we argue that it is unlikely because protease inhibitors are present in the fusion reaction buffer and should block all extra-lumenal proteolysis. Moreover, this observation is consistent with surface Hxt3-GFP being endocytosed and delivered to the membrane and lumen of vacuoles during the isolation procedure, prior to fusion in vitro (see Fig. 5a). This step of the procedure requires 60 min, which may explain why band patterns look similar to those observed at 60 min after 2-deoxyglucose treatment in vivo (Fig. 1f). This also explains why relatively high levels of lumenal GFP fluorescence (~60% of total) are observed in vacuole preparations prior to fusion (Fig. 6b). Thus, we are confident that the observed changes in GFP-cleavage reflect lumenal Hxt3-GFP degradation that correlates with internalization during vacuole fusion, suggesting that the ILF pathway mediates Hxt3-GFP degradation in vitro.

Importantly, both heat stress and CHX significantly increased the amount of Hxt3-GFP present in the boundary (Fig. 5b, c), internalized (Fig. 5d) and degraded (Fig. 5e) by the ILF pathway in vitro. We made similar observations when experiments were repeated with vacuoles isolated from $vps36\Delta$ cells (Fig. 5b–e), confirming that ESCRTs are not required for protein sorting. However, these responses were not as robust as those observed in vivo (see Figs. 3 and 4). We reasoned that this is because Hxt3-GFP found on isolated vacuole membranes is already marked for degradation, as its presence is the product of downregulation presumably triggered by glucose withdrawal during the organelle isolation procedure (Fig. 5a) and it is degraded during fusion in vitro under control (unstimulated) conditions (Fig. 5b–e). Thus, heat stress or CHX further enhances clearance of Hxt3-GFP that is already destined for degradation. Because these responses are additive, we reasoned that heat stress and CHX likely target distinct mechanisms, at least in part, from those that respond to glucose withdrawal. Because heat stress and CHX trigger Hxt3-GFP degradation in vitro, the underlying machinery must be present on vacuole membranes, whereas the machinery that senses glucose withdrawal must be, in part, present on the plasma membrane to accommodate endocytosis, and neither likely include ESCRTs.

Next, we pretreated in vitro vacuole fusion reactions with Ypt7 inhibitors and found that they blocked Hxt3-GFP sorting, internalization and degradation (Fig. 5b–e), implicating Ypt7 and its effectors in Hxt3-GFP sorting into boundary membranes.

However, closer examination of western blots revealed that some residual Hxt3-GFP cleavage occurred in the presence of inhibitors under all conditions tested (Fig. 5e; compare 0–120 min with fusion inhibitors). One explanation is that perhaps some vacuoles were docked, and contained Hxt3-GFP in their boundary membranes, upon isolation (i.e., they were engaged in the fusion process in live cells before lysis). If so, then Ypt7 inhibitors would be unable to block subsequent Ypt7-independent stages of fusion responsible for Hxt3-GFP internalization and degradation, accounting for the observed residual GFP-cleavage. This also accounts for the small fraction of the vacuole population that contains relatively high levels of Hxt3-GFP fluorescence in the boundary and lumen in the presence of Ypt7 inhibitors (Fig. 5c, d). Thus, all things considered, we are confident that Ypt7 and the ILF pathway machinery located on the vacuole membrane is likely responsible for Hxt3-GFP degradation triggered by diverse stimuli.

Clearance of ESCRT-client proteins from endosome membranes is thought to require multiple rounds of ILV formation[3], raising the question: How efficient is protein clearance from vacuole membranes by the ILF pathway? To answer it, we initiated homotypic vacuole fusion in vitro with ATP and imaged reactions every 30 min over 2 h (Fig. 6a). Some GFP fluorescence was observed in the vacuole lumen prior to fusion (0 min) and this signal increased over the course of the fusion reaction (Fig. 6b). Moreover, the amount of Hxt3-GFP on vacuole membranes decreased over time (Fig. 6c), confirming that this transporter was getting internalized into the lumen during the fusion reaction. Similar observations were made after isolated vacuole were subjected to heat stress prior to initiation of fusion (Fig. 6a–c), except the effect was more pronounced whereby nearly all of Hxt3-GFP on the membrane was internalized into the lumen after 120 min (also see Fig. 6d), consistent with complete degradation observed by western blot analysis (Fig. 5e). Previously, we showed that heat stress has no effect on the rate of vacuole fusion in vitro[37] and it is known that the population of isolated vacuoles undergoes 1.5 fusion events after 120 min under these conditions[46]. Thus, we estimate that Hxt3-GFP can be completely cleared from vacuole membranes within two fusion events.

**ILF pathway mediates quality control of many surface proteins**. Is the ILF pathway responsible for downregulation of other surface transporter proteins? To answer this (second) question, we used fluorescence microscopy to screen for other ILF-client proteins by examining the intracellular membrane distribution of GFP-tagged surface transporter or receptor proteins after live cells were challenged with heat stress, which should trigger degradation of all surface polytopic proteins. Five surface proteins showed clear phenotypes: Unlike Can1, Ste3 (a G-protein coupled receptor) or Mup1 (a methionine permease [see [47]]), internalized GFP-tagged Itr1 (a myo-inositol transporter)[40] and Aqr1 (a major facilitator superfamily-type transporter that excretes amino acids)[48] appeared on vacuole membranes on route to the vacuole lumen for degradation (Fig. 7a, c). Both transporters were present within boundary membranes between organelles (Fig. 7b) and accumulated in the vacuole lumen of live wild type cells after heat stress (Fig. 7d). This correlated with enhanced Itr1-GFP and Aqr1-GFP degradation after heat stress as assessed by western blot analysis of whole-cell lysates (Fig. 7e). Importantly, deleting VPS27, a component of ESCRT-0, had no effect on the, internalization, sorting or degradation of Itr1-GFP or Aqr1-GFP after heat stress (Fig. 7a–e), confirming that the ESCRT-independent ILF pathway is critical for quality control of these transporter proteins. Thus, the ILF pathway mediated degradation of three of

six proteins examined, suggesting that it may play a considerable role in surface transporter protein downregulation.

## Discussion

Here we demonstrate that whereas some internalized surface polytopic proteins (Can1) rely on the canonical ESCRT-dependent MVB pathway for downregulation, other presumed ESCRT-client proteins (Hxt3, Itr1, Aqr1) bypass ESCRT function at endosomes and, upon MVB-vacuole fusion, are delivered to vacuole membranes where they are selectively sorted for degradation by the ILF pathway (see Fig. 1a). Both pathways are triggered by the presence of toxic substrates, substrate withdrawal or cycloheximide and are critical for protein quality control. Thus, the assumption that all surface proteins are degraded by the MVB pathway is no longer reasonable, and we speculate that the ILF pathway may play an equally important role in surface polytopic protein downregulation. This conclusion raises many new questions concerning this important area of fundamental cell biology: What evidence supports ESCRT-mediated protein downregulation? What distinguishes one pathway from the other? What determines protein entry into each pathway? What is the potential impact of this discovery on eukaryotic physiology?

Herein, we reveal that internalized surface polytopic proteins may take two distinct pathways to the vacuole lumen for degradation. Paramount to this discovery was visualizing internalized surface receptors within live cells over time to track their routes to the vacuole lumen for degradation (e.g., Figure 1c). This revealed that some internalized proteins accumulate on the vacuole membrane: a trafficking intermediate that does not occur in the MVB pathway, but is a requirement for the ILF pathway. Unfortunately, the majority of previous reports on surface protein degradation do not provide such detailed kinetic analyses and instead present only light micrographs of cells before and after treatment, indicating surface and vacuole lumen protein distributions respectively[13,14,41,49–52]. Many important papers originally describing degradation of surface receptors or transporters do not include any micrographs[53]. Thus, the pathway responsible for their degradation now seems enigmatic. Moreover, when comprehensive datasets are provided, many internalized surface transporters appear on vacuole membranes[38], suggesting that ILF-dependent degradation of surface proteins may be widespread.

Further confounding this issue is that protein sorting and ILV formation by ESCRTs cannot be accurately visualized in biological systems using light microscopy due to limitations of spatial resolution. Rather, samples must be fixed and imaged using electron microscopy coupled with immune-gold labeling. Due to associated technical challenges, very few reports provide electron micrographs to definitively demonstrate protein sorting or internalization by ESCRTs[54–56]. However, for the ILF pathway, here we show surface transporters being delivered to the lumen during homotypic vacuole fusion in real time within live cells using HILO microscopy (Figs. 2f, 3e, and 4e; Supplementary Movies 1–5). In addition to providing definitive proof that the ILF pathway contributes to degradation of these transporters, this approach (including FRAP[37]) offers the advantage of allowing us to study the fundamental properties of protein sorting for transporter and receptor downregulation.

Finally, although we present data from complimentary experimental approaches to support our conclusions, we recognize that both pathways rely on the same membrane trafficking route within cells, i.e. endocytosis culminating with MVB-vacuole fusion (see Fig. 1a). As such, we expect that genetic manipulations affecting both pathways will cause pleiotropic phenotypes. For example, in addition to blocking ILV formation, deleting ESCRT

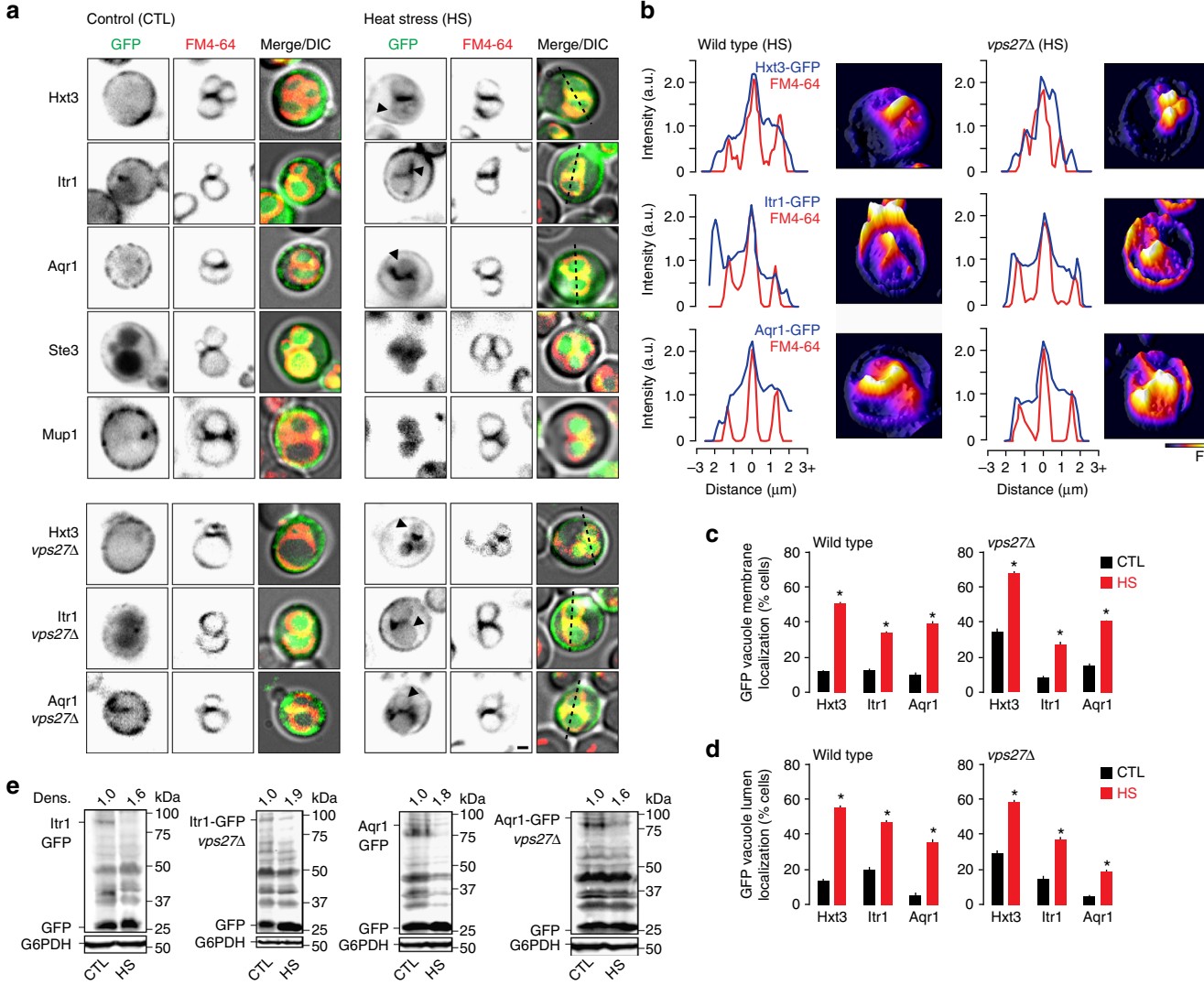

**Fig. 7** Quality control of surface transporters Itr1 and Aqr1 is mediated by the ILF pathway. **a** Fluorescence and DIC micrographs of live wild type (top) and *vps27Δ* (bottom) cells expressing GFP-tagged Hxt3, Itr1, Aqr1, Ste3, or Mup1 before (control) and after heat stress (37 °C for 15 min). Vacuole membranes were stained with FM4–64. Arrowheads indicate GFP on the vacuole membrane. Scale bar, 1 μm. **b** Three-dimensional GFP fluorescence intensity (FI) plots and line plots (left) of GFP or FM4–64 fluorescence intensity for lines shown in **a** to indicate boundary membrane localization after heat stress. **c, d** Using micrographic data shown in **a**, we measured the proportion of wild type or *vps27Δ* cells with GFP fluorescence observed on the vacuole membrane (**c**) or within the lumen (**d**) before (CTL) or after heat stress. Averages ± S.E.M. are shown. *P < 0.05, as compared to CTL by *t*-test. *n* = 4 experiments whereby a total of 238 Hxt3-GFP cells, 180 Itr1-GFP cells, 180 Aqr1-GFP cells were analyzed under CTL conditions, and 222 Hxt3-GFP cells, 161 Itr1-GFP, 167 Aqr1-GFP cells were analyzed after HS. **e** Western blot analysis of whole-cell lysates prepared from wild type or *vps27Δ* cells expressing Itr1-GFP or Aqr1-GFP before (CTL) or after heat stress (HS) stained with anti-GFP antibody. Estimated molecular weights and cleaved GFP band densities (Dens.) relative to CTL normalized to load controls (G6PDH) are shown. Blots are representatives of *n* = 3 experiments

genes partially inhibits MVB-vacuole fusion[57], which is required to deliver internalized proteins to the ILF pathway, and prevents efficient delivery of some biosynthetic cargoes to the vacuole lumen (e.g., carboxypeptidase S and other vacuole hydrolases)[3]. This perhaps explains why Hxt3-GFP accumulates on puncta and the vacuole membrane in *vps36Δ* cells (e.g., Fig. 1c–e) and its degradation profile is different than in wild-type cells (Figs. 1f, 3f, 4f, 5e, and 7e). If true, then appearance of internalized proteins on these aberrant structures in ESCRT knockout cells is not a phenotype exclusive to ESCRT-client proteins, warranting reevaluation of published data that relies on this phenotype to infer protein degradation by the MVB pathway. All things considered, we provide a logical framework to determine which pathway mediates surface protein degradation and are currently conducting screens to comprehensively understand the contributions

of each to surface receptor and transporter down-regulation in *S. cerevisiae*.

From a mechanistic perspective, both pathways are processes that share four general stages required for selective protein degradation:[3,37] (1, labeling) The protein must be labeled for degradation. (2, sorting) Labeled proteins must be selectively sorted into an area of membrane that is (3, internalization) internalized as a vesicle (or membrane fragment) into the lumen of an endocytic organelle. (4, degradation) Protein-laden lumenal vesicles must be exposed to vacuolar hydrolases for catabolism. Given these functional similarities, we argue that the underlying molecular machinery must, in part, be shared by both pathways. However, it must also differ to account for their distinctions.

Although little is known about protein labeling in the ILF pathway, we propose that the underlying mechanisms are likely

shared with the MVB pathway, which employs protein ubiquity-lation as a label for selective degradation[58]. This is based on the following reasoning: (1) prior to this work, Hxt3, Itr1 and Aqr1 were reported to be ubiquitylated by specific E3-ligase and E4-adapter proteins in response to the same stimuli that drive their downregulation by the ILF pathway (e.g., Rsp5 and Art5, respectively, for Itr1)[40]. 2 The same ubiquitylation machinery labels polytopic proteins at the plasma membrane, endosome membranes, and vacuole membranes (e.g., E3 ligases Rsp5 and Pib1, and E4 adapter Ssh4)[40,43,59]. This possibly explains why heat stress triggers degradation of Hxt3-GFP on isolated vacuole membranes in vitro (Figs. 5 and 6), as is does for Hxt3-GFP on the plasma membrane in live cells (Figs. 3 and 7). 3 When ESCRT genes are deleted, ESCRT-client proteins Can1, Ste3, and Mup1 continue to be degraded, albeit less efficiently, by the ILF pathway[47], suggesting that the machinery underlying sorting in each pathway recognizes a common label. Furthermore, the ubiquitylation machinery mediates selective soluble protein degradation by other cellular pathways (e.g., Rsp5, Ydj1, and the proteasome)[60]. Sharing the same sensing and labeling machinery permits harmonization of protein degradation pathways, including the ILF and MVB pathways, for coordinated compartmental changes in proteostasis needed for a global cellular response to a single stimulus and this redundancy also ensures toxic proteins are destroyed.

It is clear from this study that the mechanisms underlying protein sorting into each pathway are distinct: ESCRTs drive sorting in the MVB pathway[3,58], an unknown, ESCRT-independent mechanism that requires Rab-GTPase activation is important for sorting in the ILF pathway (Fig. 5) [see [37]]. Location of the sorting machinery is not likely critical, as ESCRTs are found on endosome and vacuole membranes[43], and the machinery thought to underlie protein sorting into ILFs is present on endosome and vacuole membranes where they drive fusion events (e.g., Ypt7)[57,61]. The mechanisms underlying internalization are distinct as well: In the MVB pathway, labeled proteins are internalized by ILV formation culminating with lipid bilayer scission by ESCRT-III and the Vps4-Vta1 complex on the endosome membrane[62]. Whereas, labeled proteins are internalized by ILF formation during bilayer fusion by fusogenic proteins (SNAREs) and lipids concentrated at the vertex ring formed between two docked vacuoles[33–35,37]. Finally, the mechanisms underlying degradation are essentially shared as the outcomes of both processes are equal: polytopic proteins embedded in membrane vesicles (or fragments) are degraded by acid hydrolases with the vacuole lumen[3,37].

In terms of efficiency, each ILV generated by ESCRTs has an estimated membrane area of $0.03\mu m^2$ (assuming it is spherical with a radius of $0.05\,\mu m$[63]) whereas each ILF generated by a vacuole fusion event has an area up to $2.26\,\mu m^2$ (assuming it is a double boundary membrane disk with a radius of $0.6\mu m$, average vacuole radius of $1.0\,\mu m$, and all of the boundary membrane is internalized)[36]. Thus, membrane internalized by a single ILF is equivalent to 75 ILVs. Given that each MVB contains approximately 30 ILVs[63], at least three MVB-vacuole fusion events are necessary to deliver the same amount of protein-laden membrane to the vacuole lumen as a ILF produced by one vacuole fusion event. Herein, we also show that nearly all Hxt3-GFP is cleared from vacuole membranes by the ILF pathway within two rounds of vacuole fusion in vitro (Fig. 6). In live cells, individual vacuole fusion events require 30–60 s (e.g., Fig. 2f and Supplementary Movie 1) under conditions that promote Hxt3-GFP enrichment within boundary membranes (Fig. 2c). Endocytosis can occur within min[64]. Thus, when considering the capacity and kinetics of this process, it is easy to envision how the ILF pathway can completely down-regulate Hxt3 in as little as 5 min, as observed after glucose withdrawal (Fig. 5a). Furthermore, this possibly

accounts for faster delivery of the ILF-client Hxt3-GFP from the surface to the vacuole lumen as compared to the ESCRT-client Can1-GFP observed in response to their cognate toxins within live cells (Fig. 1e). In all, it seems that the ILF pathway offers the cell a relatively high-capacity solution for rapid surface polytopic protein degradation.

Surface receptor and transporter protein downregulation is critical for diverse physiology in all eukaryotes. ESCRT-dependence has been demonstrated for only a handful of human receptors and transporters. These represent a miniscule fraction of an estimated ~5500 polytopic proteins encoded by the human genome[65], all of which presumably have finite lifetimes and must be selectively degraded. Thus, for the subset that represents surface polytopic proteins, it is reasonable to propose that this burden is split by at least two pathways. For example, here we discovered that the ILF pathway mediates downregulation of the yeast hexose transporter (Hxt3) when glucose is withdrawn from the growth medium (Fig. 5a). Downregulation of orthologous glucose transporters (e.g., GLUT4) triggered by glucose depletion is observed in mammalian epithelial cells lining the ileum or proximal tubule of the nephron[66–68], but it is not clear what mediates this process, which is needed for sugar (re)absorption. Genes encoding the proposed machinery underlying the ILF pathway are conserved in all eukaryotes including humans[69], like genes encoding ESCRTs[3]. Thus, it is tempting to speculate that the ILF pathway also contributes to the down-regulation of these sugar transporters, as well as other receptors and transporters, in mammalian cells.

Why two pathways? Evolutionary theory predicts that each pathway must perform a distinct, fundamental function necessary for eukaryotic physiology to be conserved. The concept of two pathways for surface polytopic protein downregulation is obviously still in its infancy, but in support, we offer three speculative explanations for their coexistence: (1) whereas, the function of ESCRT clients, such as Can1, may be unnecessary or not tolerated on vacuole membranes, ILF-client proteins may function on vacuoles (or lysosomes) as well as on the plasma membrane. However, this is unlikely for Hxt3 because after it is endocytosed, it concentrates in boundary membranes (Figs. 2c, 3c, 4c, and 5c) and is efficiently cleared from vacuole membranes (see Fig. 6). (2) Unlike ILF clients, ESCRT-client proteins may require immediate sequestration from the cytoplasm after endocytosis. For example, activated hormone receptors must be sequestered into ILVs for accurate signal termination[8]. (3) ESCRT clients are can be reused either by ILV back fusion, i.e., membrane fusion between ILV and MVB perimeter membranes (although this is contentious) or by MVB-plasma membrane fusion which releases protein-laden ILVs as exosomes[8]. Whereas degradation is certain for ILF-client proteins.

In addition to their role in the MVB pathway, ESCRTs are recruited to other compartmental membranes where they make important contributions to cytokinesis, reformation of the nuclear envelope, and exosome release—process that terminate with membrane fission[70,71]. Orthologous components of the ILF machinery (e.g., CORVET) are present on other endosomal compartments where it drives fusion events needed for membrane trafficking as well as organelle biogenesis and homeostasis[61,69]. Thus, from a broader perspective, it seems that ESCRTs drive membrane remodeling prior to fission, whereas the components of the ILF pathway remodel membranes prior to fusion within eukaryotic cells. Through these different mechanisms, we speculate that the ILF pathway plays a distinct but equally important role as the MVB pathway in surface transporter and receptor downregulation.

**Table 1 Yeast strains used in this study**

| Strain | Genotype | Source |
|---|---|---|
| BY4741 | *MATα his3-Δ1 leu2-Δ0 met15-Δ0 ura3-Δ0* | 77 |
| Can1-GFP | BY4741, Can1-GFP::HIS3MX | 77 |
| Ste3-GFP | SEY6210, Ste3-GFP::KanMX | 78 |
| Mup1-GFP | SEY6210, Mup1-GFP::KanMX | 78 |
| Hxt3-GFP | BY4741, Hxt3-GFP::His3MX | 77 |
| Hxt3-GFP:vps36Δ | BY4741, Hxt3-GFP::His3MX, vps36Δ::KanMX | This study |
| Hxt3-GFP:vps27Δ | BY4741, Hxt3-GFP::His3MX, vps27Δ::KanMX | This study |
| Itr1-GFP | BY4741, Itr1-GFP::His3MX | 77 |
| Itr1-GFP:vps27Δ | BY4741, Itr1-GFP::His3MX, vps27Δ::KanMX | This study |
| Aqr1-GFP | BY4741, Aqr1-GFP::His3MX | 77 |
| Aqr1-GFP:vps27Δ | BY4741, Aqr1-GFP::His3MX, vps27Δ::KanMX | This study |
| Fet5-GFP | BY4741, Fet5-GFP::HIS3MX | 77 |
| Fth1-GFP | BY4741, Fth1-GFP::HIS3MX | 77 |
| Vph1-GFP | BY4741, Vph1-GFP::HIS3MX | 77 |

## Methods

**Yeast strains and reagents**. All *Saccharomyces cerevisiae* strains used in this study are listed in Table 1. All unique yeast strains generated for this study are available from the authors. To knock out *VPS27* by homologous recombination we used the Longtine method[72] and the following primers: (forward; CBO202) 5′-GCTAAGGTGAATGAGTAGTGAGTAAAGAACTAAGAACAGTCGGAT CCCCGGGTTAATTAA-3′ and (reverse; CBO203) 5′-CTAGGTTCCTTTTTA CAAATACATAG AAAAGGCTACAATAGAATTCGAGCTCGTTTAAAC-3′; and extended the homologous regions of genomic DNA (underlined) covering 80 base pairs to improve targeting and integration using the following primer set: (forward; CBO204) 5′-TAGAGGGTGTAAAATTATCAAG ATTTTTTTTTGC TAAGGTGAATGAGTAGTC-3′ and (reverse; CBO205) 5′-CAAGCAATTA TATA TATATGTATGTATATATTTATAAGCGCTAGGTTTCTTTTTA CAAAT-3′. To knockout *VPS36*, we used the same approach with the following primer sets: (forward, CBO594) 5′-AGGAAGTG TGTTTTGAAAGTCATTCT TTTTTTTTTCAAAAGCGGATCCCCGGGTTAATTAA-3′ and (reverse; CBO596) 5′-ATGTAGTATTACGAGCAGGTAATCAAACCATGCATTTATTGAATTC GAGCTCGTT TAAAC-3′, as well as (forward, CBO595) 5′-AGGTCATCCAAG ATAGAATAGGGGACC TCGGCTAGGAATCAGGAAGTGTGTTTTGAAAGT-3′ and (reverse; CBO597) 5′-TTTAGCC GCGCGTTTATATGTAAACTATTTA TTTGGGAGAATGTAGTATTACGAGCAGGT-3′. Genomic mutations were confirmed by PCR and sequencing.

Biochemical and yeast growth reagents were purchased from either Sigma-Aldrich, Invitrogen, or BioShop Canada Inc. Proteins used include recombinant Gdi1 purified from bacterial cells using a calmodulin-binding peptide intein fusion system[73] and recombinant Gyp1–46 (the catalytic domain of the Rab-GTPase activating protein Gyp1) purified as previously described[74]. Reagents used in fusion reactions were prepared in 20 mM Pipes-KOH, pH 6.8, and 200 mM sorbitol (Pipes Sorbitol buffer, PS).

**Yeast cell viability assay**. To assess cell viability, yeast were cultured in SC medium for 16–18 h, pelleted, and resuspended in fresh SC medium. Cells were then treated with 0, 0.2 or 2 mM 2-deoxyglucose and incubated at 30 °C for 2 h, or with 0, 37, or 340.6 μM canavanine and incubated at 30 °C for 8 h. After incubation, cells were washed once with SC medium, resuspended in 100 μL SC containing 0.1% (w/v) methylene blue and incubated at room temperature for 5 min prior to imaging. Micrographs were acquired using a Nikon Eclipse TiE inverted epifluorescence microscope equipped with a Nikon DsRi2 high resolution (4908 × 3264 pixels) color CMOS (Complementary Metal-Oxide Semiconductor) camera, a Nikon CFI × 40 Plan Apo Lambda 0.95 NA objective lens and DIC optics. Methylene blue positive (MB + ) and unstained cells were counted manually using ImageJ software (National Institutes of Health, USA).

**Highly inclined laminated optical sheet (HILO) microscopy**. Micrographs were acquired using a Nikon Eclipse TiE inverted microscope equipped with a motorized TIRF (Total Internal Reflection Fluorescence) illumination unit, Photometrics Evolve 512 EMCCD (Electron Multiplying Charge Coupled Device) camera, Nikon CFI ApoTIRF 1.49 NA ×100 objective lens, and 488 nm or 561 nm 50 mW solid-state lasers operated with Nikon Elements software. Cross sectional images were recorded 1 μm into the sample.

**Live cell imaging**. Live yeast cells were stained with FM4–64 to label vacuole membranes using a pulse-chase method as previously described[45]. To trigger downregulation of GFP-tagged surface polytopic proteins, cells in SC medium were either treated with 0.2 mM 2-deoxyglucose (Sigma-Aldrich) for up to 120 min at 30 °C, 340.6 μM canavanine (Sigma-Aldrich) for up to 8 h at 30 °C, 100 μM cycloheximide for 90 min at 30 °C, resuspended in SC medium without D-glucose for 5 min at 30 °°C, or incubated at 37 °C for 15 min for heat stress. After treatments, cells were washed and resuspended in SC medium at 30 °C prior to imaging by HILO microscopy. Time-lapse videos were acquired for 5 min at 30 °C using a Chamlide TC-N incubator (Live Cell Instruments) with cells plated on coverslips coated with concavalin-A (1 mg/mL in 50 mM HEPES, pH 7.5, 20 mM calcium acetate, 1 mM MnSO₄).

**Vacuole isolation and homotypic vacuole fusion**. Vacuoles were purified from yeast cells as previously described[75], whereby cells were harvested, washed, treated with oxalyticase to generate spheroplasts (shown Fig. 5a), gently permeabilized using DEAE-dextran (also shown in Fig. 5a) and then organelles were separated on a ficoll gradient by ultracentrifugation (100,000 × *g*, 90 min, 4 °C). Homotypic vacuole fusion reactions were prepared using 6 μg of isolated vacuoles in standard fusion reaction buffer with 0.125 M KCl, 5 mM MgCl₂, and 10 μM CoA. 1 mM ATP was added to trigger fusion. Vacuolar membranes were stained with 3 μM FM4–64 for 10 min at 27 °C. Reactions were incubated at 27 °°C for up to 120 min and placed on ice prior to visualization by HILO microscopy. Where indicated, vacuoles were incubated in the absence or presence of the fusion inhibitors 3.2 μM Gyp1–46 and 4 μM rGdi. For heat stress or cycloheximide treatment, isolated vacuoles were incubated at 37 °C for 5 min or at 27 °C with 100 μM cycloheximide for 15 min, respectively, prior to adding them to the fusion reaction.

**Western blot analysis**. For analysis of whole-cell lysates, yeast cells were prepared as previously described[76]. In brief, extracts were prepared by harvesting yeast cells grown in culture to mid-log phase and washing them once with YPD medium. Cells were then resuspended in fresh YPD and incubated at 30 °C in the presence of 0.2 mM 2-deoxyglucose or 340.6 μM canavanine for times indicated in Fig. 1f, at 30 °C for 90 min for control conditions, at 30 °C in the presence of 100 μM cycloheximide for 120 min, or incubated at 37 °C for 90 min for heat stress. After treatment, 5 OD₆₀₀ₙₘ units of cells were collected, resuspended in 0.5 mL of lysis buffer (0.2 M NaOH, 0.2% β-mercaptoethanol) and incubated on ice for 10 min. Trichloroacetic acid was then added to a final concentration of 5% and samples were incubated on ice for an additional 10 min. Precipitates were collected by centrifugation (12,000 × *g* for 5 min at 4 °C) and resuspended in 35 μL of dissociation buffer (4% SDS, 0.1 M Tris-HCl, pH 6.8, 4 mM EDTA, 20% glycerol, 2% β-mercaptoethanol and 0.02% bromophenol blue). Tris-HCl, pH 6.8 was then added to a final concentration of 0.3 M and samples were incubated at 37 °C for 10 min. For analysis of in vitro fusion reactions, samples were prepared from isolated vacuoles as previously described[37]. Whole-cell lysates or isolated vacuole preparations were loaded into 10% SDS-polyacrylamide gels and after electrophoresis separated proteins were transferred to nitrocellulose membranes. Membranes were then probed with antibodies raised against GFP (1:1,000; B2, Santa Cruz Biotechnology Inc., Cat# sc-9996; or Abcam, Cat# ab290), G6PDH (1:1000; Sigma-Aldrich, Cat# A9521) or Pho8 (1:1000; 1D3A10, Abcam, Cat# ab113688). Horse-radish peroxidase-labeled affinity purified polyclonal antibodies to either rabbit or mouse (1:10,000; SeraCare, Cat# 5450-0010 or 5450-0011) were used for secondary labeling. Please refer to manufacturer's website for antibody validation. Chemiluminescence of stained membranes was digitally imaged using a GE Amersham Imager 600.

**Data analysis and presentation**. Micrographs and movies were processed using ImageJ and Adobe Photoshop CC software. Images shown were adjusted for brightness and contrast, inverted and sharpened with an unsharp masking filter. Linear or 3-dimnesional fluorescence intensity profiles of GFP or FM4–64 fluorescence were generated using ImageJ software. Snapshots from movies were selected to highlight docking and ILF formation during vacuole fusion. Group allocations were blinded for all micrograph analysis.

GFP location measurements shown in Figs. 1e, 2e, 3b, 4b, 7c, d were generated using the ImageJ Cell Counter plugin. Micrographs were quantified by counting the total number of cells, as well as the number of cells where the GFP fluorescence was detected on the plasma membrane (PM), intracellular puncta, the vacuole membrane (Vac mem), or in the vacuole lumen. Values for each cellular GFP localization were normalized to the total number of cells. In Fig. 2g, movies vacuole fusion events in vivo were used to determine the percent of fusion events; Only cells containing two or more vacuoles were counted.

Relative vacuole boundary, lumenal or outside membrane GFP fluorescence values were measured using ImageJ software [see [37]]. Prior to quantification, micrographs were background subtracted and a 4 × 4 pixel region of interest was then used to measure the mean GFP fluorescence within the boundary, lumen or on the outer membrane of docked vacuoles only. Single vacuoles or docked vacuoles without a clear outer membrane (i.e., those in large clusters) were excluded. Mean GFP fluorescence intensity was correlated to vacuole size (boundary length, surface area or circumference) by measuring the vacuole

diameter (average of two lengthwise measurements) and the length of the boundary membrane. Boundary GFP fluorescence is normalized to that of the outer membrane and lumenal GFP fluorescence is normalized to total fluorescence (boundary, outer membrane and lumen). Only cells containing clearly resolved docked vacuole membranes were used for micrographic analysis.

When applicable, data are reported as cumulative probability plots as well as mean ± S.E.M. Comparisons were calculated using Student two-tailed $t$-test; $P$ values < 0.05 (*) indicate significant differences. An experiment is defined as a sample prepared from a separate yeast culture on different days. The total number of cells, vacuoles, or boundaries analyzed are indicated. Samples sizes were tested to ensure adequate power using online software (http://www.biomath.info). Variance was assumed to be similar between groups compared. Data was plotted using Synergy KaleidaGraph 4.0 software and figures were prepared using Adobe Illustrator CC software.

## Data availability

Data that support findings of this study are available from the corresponding author upon reasonable request.

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

## Acknowledgements

We thank Beverley Wendland and Derek Prosser for yeast strains. We thank the Canadian Foundation for Innovation and Natural Sciences and Engineering Research Council of Canada for generous support of the Centre for Microscopy and Cellular Imaging at Concordia University. This work was supported by NSERC grants RGPIN/403537-2011 and RGPIN/2017-06652 to C.L.B.

## Author contributions

C.L.B and E.K.M. conceived the project. E.K.M. performed experiments and prepared all data for publication. C.L.B. guided experimental design and analysis. E.K.M. and C.L.B. wrote the paper.

## Additional information

**Competing interests:** The authors declare no competing interests.

