## [Peer Review File · Nature Communications]

Reviewers' comments:

Reviewer #1 (Remarks to the Author):

This study has discovered a novel pathway of endocytosis for degradation of plasma-membrane proteins in the budding yeast *Saccharomyces cerevisiae*, through fusion of apposing vacuolar membranes at the edge ring, termed Intraluminal Fragment Pathway. This pathway, without utilizing ESCRT proteins, contributes to turnover processes of multiple proteins (Hxt3, Ste3, Mup1, Itr1, and Aqr1), in response to metabolic or other stresses that are thought to affect the folding states of the target proteins. The turnover process was reproduced in vitro with isolated vacuolar membranes, adding verification of functional requirement of the membrane-fusion machinery in this process. Furthermore, the authors provide valuable discussions regarding the biological significance of the dual (MVB and ILF) pathways for endocytic turnover of membrane proteins. In combination with the authors' previous studies on the ILF pathway, this study persuasively demonstrated the novel endocytic pathway with wide readership. Yet, I have to refer to three points to be overcome or clarified before the publication.

1. Induction points of Hxt3-GFP turnover: As judged by Fig. 2A, internalization of Hxt3-GFP (from the plasma membrane) in response to 2-deoxyglucose is evident. In contrast, it seems difficult to state (based on Fig. 2G) that the 2-deoxyglucose treatment alter the mode of vacuolar membrane fusion from one without the cutoff of the boundary plane to one producing ILFs, since the percentages of fusion events producing ILFs are about 94% and 97% before and after the 2-deoxyglucose treatment, given that the vacuolar membrane fusion event comprises the two modes. Hence it might be more reasonable to omit Fig. 2G, and the authors focus on the induction of internalization step in Fig. 2.

2. To clearly show the 'sorting' of Hxt3 or other plasma-membrane proteins to the boundary region within the vacuolar membrane, the authors should show images of the cells challenged by 2-deoxyglucose or other stresses that have single vacuoles. If docking of the vacuolar membrane indeed triggers the sorting upon formation of the boundary region, the fluorescent protein should be distributed uniformly in the single vacuoles.

3. Data from biochemical analyses detecting degradation of the GFP-tagged proteins seem to be inconsistent at several points. For instance, in Fig. 1F, the cleaved GFP moiety was barely detected at time 0 in all the three panels, whereas in Fig. 2F, it was detected to the same extent as the non-cleaved form (in the sample from the wild type cells) and more heavily in the *vps36d* sample. And in the in vitro experiments (Fig. 5E), the cleaved form was detected in almost all the lanes but one (wild type, control to the cycloheximide treatment, time 0). This maybe arises from the degradation of the transporters in the lysate acquisition step. It is empirically known that some yeast peptidases are resistant to SDS treatment. Suspension in the dissociation buffer can lead to artificial degradation of the proteins. I would recommend the authors to include some protease inhibitors in the dissociation buffer, and repeat at least the experiment for Fig. 2F.

Reviewer #2 (Remarks to the Author):

In this paper, McNally and Brett employ a yeast system to assess the mechanism by which surface transporters are degraded by the lysosome-like vacuole upon toxin-stimulated downregulation. Triggered surface receptors in yeast (and mammals) are thought to be degraded largely following ubiquitylation and recognition by the ESCRT machinery on endosomes, which target the receptors to intraluminal vesicles that are then degraded following late endosome fusion with the vacuole. In a phenomenal 2017 paper in *Dev. Cell*, these authors identified a novel mechanism for degradation of lysosomal membrane proteins by the "intraluminal fragment" (ILF) pathway, by which they are incorporated into boundary membranes during vacuole: vacuole fusion and degraded upon release of these membranes into the vacuole lumen. Here, in an equally rigorous study, the authors show that surface receptors can be degraded by the same mechanism, focusing primarily on the glucose transporter Hxt3 but supported by two additional receptors in the final

figure. This extends the relevance of the ILF pathway to not only degradation of damaged lysosomal proteins but also to physiological down-regulation of a subset of surface receptors.

The paper addresses an important question in the field. It challenges the prevailing notion that the ESCRT pathway is responsible for all surface receptor down-regulation, and does so quite convincingly - including by confirming that some receptors are indeed degraded by the ESCRT pathway. It extends the authors' previous study on lysosomal protein degradation in an important way, and thus seems appropriate for the readership of *Nat. Commun.* Like the 2017 paper, the study is extremely thorough, nicely quantitative, and covers all of their bases with the experimental design. The conclusions are nicely supported by the data, and the discussion of the results is scholarly and thoughtful. Main concerns are relatively minor, relating to concerns over a few of the many panels shown in the paper and interpretation of some of the experiments (particularly the cleavage data, which do not support the authors' conclusions from the experiment but likely do not affect the main conclusion), with really only one additional suggested experiment below. I also suggest judicious editing of the Discussion and limiting the discussion to points validated in the paper.

1. In Figure 1C, the localization of Hxt3-GFP in wild-type cells at the 5' time point is not terribly obvious; it might be best to replace this panel with a more representative image. The authors should comment on the increased accumulation of Hxt3-GFP on the vacuolar membrane in *vps36delta* cells relative to wild-type cells in Figure 1E. The authors should also comment on the different kinetics and efficiency of GFP cleavage from Can1 or Hxt3 shown in Figure 1F. If extended longer, is GFP cleaved from Hxt3 more at later stages, like from Can1? Lastly, this figure would benefit from showing the impact of *vps36delta* on Can1-GFP as a control for the more modest effects on Hxt3-GFP.

2. The arrowhead in Figure 2A does not point to Hxt3-GFP staining. In interpreting the data in Figure 2D, the authors should be cautious in drawing conclusions regarding turnover and down-regulation since this is not measured, and observations are limited to single cells in each panel. In Figure 2E and Movie S1, the accumulation of Hxt3-GFP in boundary membranes is not obvious; perhaps a better example might be shown (by contrast, it is very obvious and totally cool in Figures 3E and 4E and their corresponding movies). In Figure 2F, it would be helpful to show examples of fusion events with and without ILFs to support the quantitative data, and the presentation of these data as "fusion events without ILF formation" is confusing; showing the data in a more positive light for 2DG treatment would be clearer for readers.

3. The arrowhead in Figure 3E, top does not point to the ILF in this image. In Figure 3F, any increase in free GFP after heat shock in *Vps36delta* cells is not obvious. It would be helpful if the blots were quantified, with free GFP relative to a constant band from the same blot. If this is not possible, at least a more representative blot should be shown. The same is true in Figure 4F. Perhaps degradation is impaired in *Vps36delta* cells due to the impaired sorting of vacuolar hydrolases? Finally, the response of this substrate to treatment with cycloheximide in Figure 4 cannot be attributed to TOR signaling without repeating the nice controls included in the 2017 *Dev. Cell* paper for other substrates.

4. The subheading on Page 9 seems to apply to the previous figures. It would be best to add "in vitro" at the end to distinguish this section from the previous sections.

5. In Figure 5B, the accumulation of Hxt3-GFP at the boundary in the cell treated with CHX is not at all obvious. Also, the quantitative impact of CHX or HS on boundary localization in Figure 5C and on degradation in Figure 5E are not so obvious. This should be admitted in the text, and likely reflects the already robust sorting and cleavage in the in vitro assay induced by the spheroplast procedure; i.e., this does not affect the main conclusions of the paper. Again, degradation of Hxt3-GFP in *Vps36delta* cells, particularly in the left-hand panel of Figure 5E, is not at all impressive and should be qualified in the text. The effects of the Ypt7 inhibitors on degradation appear to be quite

variable (e.g. HS still increases degradation in the last lanes of the left bottom panel of Figure 5E and CHX still somewhat increases degradation in the last lanes of the right top panel). This should be qualified or better explained.

6. The only additional experiment that should be done is an in vivo experiment with heat shock or cycloheximide treatment to complement the in vivo assay shown in Figure 6 to assess the efficiency of Hxt3-GFP clearance from the plasma membrane and sorting into the lumen of the vacuole.

7. In Figure 7, again, while the sorting of Aqr1-GFP and Itr1-GFP into vacuole membranes, boundary membranes and the vacuole lumen in a vps27-independent way in panels A-D is obvious, the degradation upon heat shock in panel E is not. Itr1-GFP is barely degraded at all in wild-type cells, and HS-dependent degradation of Aqr1 is eliminated by vps27 deletion. These data do not support the authors' conclusions regarding degradation. Note, the Figure legend refers at one point to vps36delta cells, whereas in this experiment vps27delta cells are used.

8. Following points in the discussion need to be addressed:

- generally, the discussion is highly speculative and reads more like a commentary than a discussion of the contextual placement of the data in the paper within the field. I would recommend trimming the discussion substantially to focus on the contribution this paper makes rather than speculation on unaddressed issues, such as sorting into the ILF pathway, roles of lipids or ubiquitylation, etc. These are interesting points, but inappropriate for a discussion of this paper.
- page 14, top: super-resolution microscopy allows distinction between intraluminal vesicles and the lysosomal limiting membrane, including STED which can be used for live cell microscopy.
- page 14, bottom: re-evaluation of the GFP release data in this paper supports the contention raised here that disabling the ESCRT pathway can interfere with vacuolar/ lysosomal degradation.

RESPONSE TO REVIEWERS

First, we would like to thank both reviewers for their insightful and constructive comments. We addressed them all to improve the original study. Please note that all requested changes to the text are highlighted in red in the attached revised manuscript.

5 REVIEWER #1:

*This study has discovered a novel pathway of endocytosis for degradation of plasma-membrane proteins in the budding yeast *Saccharomyces cerevisiae*, through fusion of apposing vacuolar membranes at the edge ring, termed Intraluminal Fragment Pathway. This pathway, without
10 utilizing ESCRT proteins, contributes to turnover processes of multiple proteins (*Hxt3*, *Ste3*, *Mup1*, *Itr1*, and *Aqr1*), in response to metabolic or other stresses that are thought to affect the folding states of the target proteins. The turnover process was reproduced in vitro with isolated vacuolar membranes, adding verification of functional requirement of the membrane-fusion
15 machinery in this process. Furthermore, the authors provide valuable discussions regarding the biological significance of the dual (MVB and ILF) pathways for endocytic turnover of membrane proteins.*

*In combination with the authors' previous studies on the ILF pathway, this study persuasively demonstrated the novel endocytic pathway with wide readership. Yet, I have to refer to three
20 points to be overcome or clarified before the publication.*

*1. Induction points of *Hxt3*-GFP turnover: As judged by Fig. 2A, internalization of *Hxt3*-GFP (from the plasma membrane) in response to 2-deoxyglucose is evident. In contrast, it seems difficult to state (based on Fig. 2G) that the 2-deoxyglucose treatment alter the mode of
25 vacuolar membrane fusion from one without the cutoff of the boundary plane to one producing ILFs, since the percentages of fusion events producing ILFs are about 94% and 97% before and after the 2-deoxyglucose treatment, given that the vacuolar membrane fusion event comprises the two modes. Hence it might be more reasonable to omit Fig. 2G, and the authors focus on the induction of internalization step in Fig. 2.*

30
Good point. We agree, the effect of 2-deoxyglucose on ILF formation shown in Figure 2G is relatively small, although significant. Thus, we removed Figure 2G from the revised paper and adjusted the text to only focus on protein internalization, as requested, and removed all discussion on the possible effect of 2-deoxyglucose on ILF formation, i.e. the following text in
35 the original version of the Results section:

"However, we did observe a small but significant decrease in the number of fusion events that did not produce visible ILFs (Figure 2G). This is consistent with previous work showing that ILF production during vacuole fusion can be regulated (Wang et al., 2002; Mattie et al., 2017), and

suggests that 2-deoxyglucose stimulates ILF formation to accommodate selective degradation of Hxt3-GFP.”

Was replaced with:

5

Page 8, line 29: *“Thus, vacuole fusion persists under these conditions and seems to mediate the selective degradation of Hxt3 after endocytosis.”*

10 2. To clearly show the ‘sorting’ of Hxt3 or other plasma-membrane proteins to the boundary region within the vacuolar membrane, the authors should show images of the cells challenged by 2-deoxyglucose or other stresses that have single vacuoles. If docking of the vacuolar membrane indeed triggers the sorting upon formation of the boundary region, the fluorescent protein should be distributed uniformly in the single vacuoles.

15

Good point. However, I am not certain that the fluorescent protein must be uniformly distributed on vacuole membranes prior to organelle docking to demonstrate that it is sorted into the boundary region. For example, it is possible that the proteins and lipids that compose the MVB perimeter and vacuole membranes do not entirely mix after fusion, perhaps creating microdomains within the membrane of the fusion product. Upon homotypic vacuole membrane fusion, which must follow MVB-vacuole fusion for surface protein degradation by the ILF pathway, these microdomains containing internalized surface proteins, e.g. Hxt3-GFP, would still need to be present in the boundary region to be internalized and degraded by the ILF pathway. Thus, uniform membrane protein distribution prior to organelle docking is not a requirement for (or indicator of) sorting into the ILF pathway.

20

Rather, we argue that the presence of a fluorescent protein, i.e. Hxt3-GFP, in the boundary region (along with supporting evidence showing that it decorates newly formed ILFs for example) is sufficient to claim that it is indeed degraded by the ILF pathway – the central argument of this paper. Although we acknowledge that, given other proteins (Fet5-GFP) are excluded from the boundary, some sorting mechanism must exist although currently elusive.

25

We also recognize that the possibility of nonuniformity may seem to undermine some of the results presented, e.g. the linear plots of fluorescence intensity used to demonstrate the presence of the fluorescent protein in the boundary region. However, uniformity of distribution has little impact on other supporting evidence that presents total fluorescence of vacuole membrane regions, i.e. relative boundary GFP fluorescence plots. Thus we are confident with our claim that Hxt3-GFP is enriched within boundary regions as compared to the outside membrane, in support of it being sorted into the ILF pathway.

30

We are currently conducting a separate but parallel study that specifically examines how the cells degrade ILF-client proteins in cells with single vacuoles, i.e. when they cannot be degraded by the ILF pathway, which implicates ESCRT-dependent processes, e.g. the VRED

35

pathway. In these cells, fluorescent proteins that are normally uniformly distributed on multilobed vacuoles seem to concentrate into patches or microdomains on single vacuole membranes before being packaged into budding vesicles. Thus, the proposed analysis of cells with single vacuoles to assess uniform distribution may not be the best approach to assess uniformity of internalized Hxt3-GFP on vacuole membranes as vacuole morphology seems to influence distribution of other proteins.

Finally, we are also conducting a related study that addresses this concern to better understand protein sorting into the ILF pathway and the role that specific lipid species and microdomains may play in this process. Thus, we prefer to reserve all related data for a future manuscript that will present a more comprehensive answer to this excellent question.

To address this concern, we revised the text to only imply that Hxt3-GFP and other surface proteins were sorted into the ILF pathway for down-regulation. For example:

Page 8, line 25: “These important findings ~~suggest~~ *imply* that Hxt3-GFP was selectively sorted into the ILF pathway...”

3. Data from biochemical analyses detecting degradation of the GFP-tagged proteins seem to be inconsistent at several points. For instance, in Fig. 1F, the cleaved GFP moiety was barely detected at time 0 in all the three panels, whereas in Fig. 2F, it was detected to the same extent as the non-cleaved form (in the sample from the wild type cells) and more heavily in the *vps36Δ* sample. And in the *in vitro* experiments (Fig. 5E), the cleaved form was detected in almost all the lanes but one (wild type, control to the cycloheximide treatment, time 0). This maybe arises from the degradation of the transporters in the lysate acquisition step. It is empirically known that some yeast peptidases are resistant to SDS treatment. Suspension in the dissociation buffer can lead to artificial degradation of the proteins. I would recommend the authors to include some protease inhibitors in the dissociation buffer, and repeat at least the experiment for Fig. 2F.

We agree that the *in vivo* data shown in Figure 1F and Figure 3F (mistaken for 2F) of the original paper are not entirely consistent. Thus, we repeated the requested experiment for Figure 3F and the new western blots presented show similar band patterns under control conditions as those at time = 0 shown in Figure 1F, when accounting for exposure.

Thus, to address this concern we replaced the western blot presented in Figure 3F with new results from requested experiments (n = 2 biological replicates).

Concerning the *in vitro* experiments: Figure 1F presents data from whole cell lysates, whereas Figure 5E shows data from purified vacuoles. Whole cell lysates contain residual full length Hxt3-GFP present on the plasma membrane, on endosomes on route to the vacuole and on the vacuole membrane (e.g. see Figure 1E). Whereas vacuole preparations of course contain only

full length Hxt3-GFP on vacuole membranes, i.e. a fraction of total cellular full length Hxt3-GFP. However, both preparations contain all proteolytic fragments of Hxt3-GFP found in the lumen of vacuoles. Thus, we would expect proportionally more cleaved GFP relative to full-length Hxt3-GFP in all vacuole preparations, which is what is observed in Figure 5E.

5
Moreover, time = 0 minutes for the in vitro experiments is equivalent to later time points in vivo. This is because during the vacuole isolation procedure living cells are treated with buffers devoid of glucose for spheroplasting, which requires approximately 60 minutes prior to cell rupture. During this time, Hxt3-GFP is endocytosed and sent to the membrane and lumen of
10 vacuoles within intact cells (see Figure 5A). Cells are then ruptured, and vacuoles are isolated and used for in vitro experiments. Thus, time = 0 minutes in vitro should be similar to time = 60 minutes in vivo (i.e. 60 minutes after endocytosis of Hxt3-GFP was triggered). This explains why nearly all lanes contain cleaved GFP in Figure 5E (in vitro), as all time points should have similar or more amounts of cleaved GFP as time = 60 minutes in vivo (Figure 1F).

15
It is also worth noting that we did take precautions to minimize potential extra-lumenal proteolysis that may occur due to vacuole rupture during isolation, including the addition of protease inhibitors (6.7 μ M leupeptin, 33 μ M pepstatin, 1 mM PMSF and 10.7 mM AEBSF) to all in vitro vacuole fusion reactions prior to incubation at 27 °C (as suggested) and storage of
20 isolated vacuoles on ice prior to use.

Thus, to address this concern we replaced the brief explanation for the difference in banding patterns observed in vitro versus in vivo offered in the original manuscript:

25 *“We also noted that GFP was cleaved prior to fusion (0 minutes), which correlates with the presence of GFP in the vacuole lumen (also prior to fusion; Figures 5A and 6A). This likely represents Hxt3-GFP that was completely downregulated and degraded in live cells during the vacuole isolation procedure.”*

30 With a better explanation that takes into account all possible contributing factors:

Page 10, line 32: *“However, we noted that relative high levels of cleaved GFP were found in all preparations containing isolated vacuoles, including those analyzed prior to fusion (0 minutes). It is possible that vacuole rupture and release of luminal proteases may account for this
35 observation. Although we cannot completely eliminate this possibility, we argue that it is unlikely because protease inhibitors are present in the fusion reaction buffer and should block all extra-lumenal proteolysis. Moreover, this observation is consistent with surface Hxt3-GFP being endocytosed and delivered to the membrane and lumen of vacuoles during the isolation procedure, prior to fusion in vitro (see Figure 5A). This step of the procedure requires 60
40 minutes, which may explain why band patterns look similar to those observed at 60 minutes after 2-deoxyglucose treatment in vivo (Figure 1F). This also explains why relatively high levels of luminal GFP fluorescence (~60 % of total) are observed in vacuole preparations prior to fusion (Figure 6B). Thus, we are confident that the observed changes in GFP-cleavage reflect*

luminal Hxt3-GFP degradation that correlates with internalization during vacuole fusion, suggesting that the ILF pathway mediates Hxt3-GFP degradation in vitro."

5

REVIEWER #2:

10 *In this paper, McNally and Brett employ a yeast system to assess the mechanism by which surface transporters are degraded by the lysosome-like vacuole upon toxin-stimulated downregulation. Triggered surface receptors in yeast (and mammals) are thought to be degraded largely following ubiquitylation and recognition by the ESCRT machinery on endosomes, which target the receptors to intraluminal vesicles that are then degraded following late endosome fusion with the vacuole. In a phenomenal 2017 paper in Dev. Cell, these authors identified a novel mechanism for degradation of lysosomal membrane proteins by the "intraluminal fragment" (ILF) pathway, by which they are incorporated into boundary membranes during vacuole: vacuole fusion and degraded upon release of these membranes into the vacuole lumen. Here, in an equally rigorous study, the authors show that surface receptors can be degraded by the same mechanism, focusing primarily on the glucose transporter Hxt3 but supported by two additional receptors in the final figure. This extends the relevance of the ILF pathway to not only degradation of damaged lysosomal proteins but also to physiological down-regulation of a subset of surface receptors.*

25 *The paper addresses an important question in the field. It challenges the prevailing notion that the ESCRT pathway is responsible for all surface receptor down-regulation, and does so quite convincingly - including by confirming that some receptors are indeed degraded by the ESCRT pathway. It extends the authors' previous study on lysosomal protein degradation in an important way, and thus seems appropriate for the readership of Nat. Commun. Like the 2017 paper, the study is extremely thorough, nicely quantitative, and covers all of their bases with the experimental design. The conclusions are nicely supported by the data, and the discussion of the results is scholarly and thoughtful. Main concerns are relatively minor, relating to concerns over a few of the many panels shown in the paper and interpretation of some of the experiments (particularly the cleavage data, which do not support the authors' conclusions from the experiment but likely do not affect the main conclusion), with really only one additional suggested experiment below. I also suggest judicious editing of the Discussion and limiting the discussion to points validated in the paper.*

35 *1. In Figure 1C, the localization of Hxt3-GFP in wild-type cells at the 5' time point is not terribly obvious; it might be best to replace this panel with a more representative image.*

40 *We agree and replaced this micrograph in Figure 1C of the revised manuscript.*

The authors should comment on the increased accumulation of Hxt3-GFP on the vacuolar membrane in vps36delta cells relative to wild-type cells in Figure 1E.

Good point. We added the following text to the Results in the revised paper to address this concern:

5 Page 7, line 18: *“Although Hxt3-GFP continues to be delivered to the vacuole lumen over time, we noticed that on route, Hxt3-GFP abnormally accumulated on puncta and vacuole membranes, even prior to treatment, in mutant cells. This is consistent with previous reports showing that deleting ESCRT genes impairs, but does not entirely block, endocytosis and delivery of vacuole cargo proteins in addition to completely abolishing protein sorting into ILVs (see Henne et al., 2011).”*

The authors should also comment on the different kinetics and efficiency of GFP cleavage from Can1 or Hxt3 shown in Figure 1F.

15 Good point. Actually, we did mention this in the Discussion section of the original paper in context to efficiency:

Page 17, line 26: *“... Furthermore, this possibly accounts for faster delivery of the ILF-client Hxt3-GFP from the surface to the vacuole lumen as compared to the ESCRT-client Can1-GFP observed in response to their cognate toxins within live cells (Figure 1E).”*

If extended longer, is GFP cleaved from Hxt3 more at later stages, like from Can1?

25 Good question. Yes, we predict that extending the time period of 2-deoxyglucose treatment would likely result in more Hxt3-GFP degradation. So why did we stop the experiment at 3 hours?

30 When first designing these experiments, we decided to use similar concentrations and exposure times as previous reports (see MacGurn et al., 2011 and O'Donnell et al., 2015) to justify the comparison of experimental results. Consistent with these previous reports, we found that nearly 100% of cells contained Hxt3-GFP or Can1-GFP in the lumen after the treatment period and used this as an indication that the response was complete. Western blot analysis later revealed that nearly all of the detectable Can1-GFP (and cleaved GFP) was degraded by 10 hours, but some Hxt3-GFP and cleaved GFP remained at the end of the experiment.

35 However, after serious consideration, we decided not to repeat the entire set of Hxt3-GFP experiments shown in Figure 1C – F to include more time points as requested, because ultimately, this new dataset would not provide additional support to our central conclusions, as the existing data clearly demonstrates that GFP-cleavage (indicating proteolysis) correlates with progressive redistribution of Hxt3-GFP from the plasma membrane to puncta, vacuole membranes and vacuole lumen over time after 2-deoxyglucose addition.

Lastly, this figure would benefit from showing the impact of *vps36delta* on Can1-GFP as a control for the more modest effects on Hxt3-GFP.

5 We agree but have reserved this dataset for a second paper in preparation demonstrating that ESCRT client proteins like Can1 are rerouted to the ILF pathway for degradation when genes encoding components of the ESCRT machinery are deleted.

10 In fact, we included this data in the original version of the manuscript previously submitted twice to another journal but removed it from later versions at the request of the reviewers because it (apparently) caused confusion. Thus, we are reluctant to include this data in the revised paper and argue that a replacement is not necessary to support our conclusions, as it is established that Can1 is a ESCRT client protein (e.g. *MacGurn et al., 2011*; noting that we confirm this in Figure 1B) and Can1 is absent from vacuole membranes eliminating the possibility that it is a ILF client protein (Figure 1C – E).

15

2. The arrowhead in Figure 2A does not point to Hxt3-GFP staining.

20 Excellent point. To address this concern, we replaced the micrograph in Figure 2A and adjusted the arrowhead so that it clearly indicates Hxt3-GFP staining on vacuole membranes.

In interpreting the data in Figure 2D, the authors should be cautious in drawing conclusions regarding turnover and down-regulation since this is not measured, and observations are limited to single cells in each panel.

25

30 Agreed. We analyzed the cellular distribution of Fet5-GFP, Fth1-GFP, Vph1-GFP, Itr1-GFP, Can1-GFP and Hxt3-GFP (control) before and after 2-deoxyglucose treatment, and Hxt3-GFP after canavanine treatment, in the population of live cells studied. We found that with the exception of Hxt3-GFP cells treated with 2-deoxyglucose, our positive control, the distributions of the other proteins did not change with treatment in cell population.

Thus, to address this concern, we present these new results in Figure 2E and added the following description to the Results of the revised manuscript:

35 Line 8, line 23: *“We assessed all cells imaged, quantified these observations and confirmed that only Hxt3-GFP was down-regulated by 2-deoxyglucose but it was unaffected by canavanine (Figure 2E).”*

40 *In Figure 2E and Movie S1, the accumulation of Hxt3-GFP in boundary membranes is not obvious; perhaps a better example might be shown (by contrast, it is very obvious and totally cool in Figures 3E and 4E and their corresponding movies).*

Excellent point. As requested, we repeated this experiment and provide a better example of Hxt3-GFP internalization during homotypic fusion in Figure 2F (previously 2E) and Movie S1 in the revised paper.

5 *In Figure 2F, it would be helpful to show examples of fusion events with and without ILFs to support the quantitative data, and the presentation of these data as “fusion events without ILF formation” is confusing; showing the data in a more positive light for 2DG treatment would be clearer for readers.*

10 Reviewer #1 raised a similar concern (please refer to our response to Comment #1 above). In brief, to address this concern, we removed Figure 2G and omit the discussion on how 2-deoxyglucose may affect ILF formation. Thus, although this is a great suggestion, there is no longer a need to show fusion events without ILFs as requested.

15 3. *The arrowhead in Figure 3E, top does not point to the ILF in this image.*

In the original paper, the arrowhead was placed to indicate the potential site of membrane fusion initiation. However, using the arrowheads to indicate newly formed ILFs is a better idea.

20 Thus, to address this concern, we adjusted arrowheads to indicate newly formed ILFs in all figures showing movie snapshots, i.e. Figures 2F, 3E and 4E, and adjusted the figure legends accordingly.

25 *In Figure 3F, any increase in free GFP after heat shock in Vps36delta cells is not obvious. It would be helpful if the blots were quantified, with free GFP relative to a constant band from the same blot. If this is not possible, at least a more representative blot should be shown. The same is true in Figure 4F. Perhaps degradation is impaired in Vps36delta cells due to the impaired sorting of vacuolar hydrolases?*

30 Reviewer #1 raised a similar concern (please refer to our response to Comment #3 above). To address it, we repeated experiments shown in Figure 3F and replaced the western blots with new ones that better show an increase in cleaved GFP after wild type or *vps36Δ* cells are exposed to heat stress. The band patterns under control conditions shown in these new blots are (more) similar with ones shown in Figure 1F at time zero and ones observed under control conditions shown in Figure 4F.

35 Concerning blot quantification: For each condition shown, we first normalized cleaved GFP band density to G6PDH band density (the control that should remain constant in each sample) as suggested. Thus, our analysis is already accounting for potential sample loading variation, which is why we chose to instead provide a more representative blot for Figure 3F.

Concerning the impaired sorting of vacuolar hydrolases: To address this concern, we edited the following text in the Discussion section:

5 Page 15, line 18: *"For example, in addition to blocking ILV formation, deleting ESCRT genes partially inhibits MVB-vacuole fusion (Karim et al., 2018), which is required to deliver internalized proteins to the ILF pathway, and prevents efficient delivery of some biosynthetic cargoes to the vacuole lumen (e.g. carboxypeptidase S and other vacuole hydrolases; Henne et al., 2011). This perhaps explains why Hxt3-GFP accumulates on puncta and the vacuole membrane in vps36Δ cells (e.g. Figure 1C – E) and its degradation profile is different than in wild type cells (Figures 1F, 3F, 4F, 5E and 7E)."*

15 Finally, the response of this substrate to treatment with cycloheximide in Figure 4 cannot be attributed to TOR signaling without repeating the nice controls included in the 2017 Dev. Cell paper for other substrates.

20 Agreed. However, it is worth noting that prior to our study published in *Developmental Cell*, Scott Emr's group showed that cycloheximide stimulates TOR to drive down-regulation of many surface polytopic proteins by the ESCRT/MVB pathway, including Can1 (MacGurn et al., 2011). As TOR also mediates cycloheximide-triggered protein degradation by the ILF pathway, we reasoned that we could make this assumption. Although we agree that this assumption is not entirely valid, we decided not repeat all experiments shown in Figure 4 with rapamycin, puromycin and /or *fpr1Δ* cells. This is because implicating TOR signaling in the observed responses to cycloheximide does not add additional support to the central conclusion of this study which is that the ILF pathway mediates surface polytopic protein down-regulation triggered by multiple stimuli.

30 Thus, to address this concern, we adjusted the text to no longer imply that cycloheximide triggers TOR signaling in this study. For example, we replaced "TOR" with "cycloheximide" in the Summary, Highlights and eTOC Blurb, Graphical Abstract and Discussion sections of the revised paper. However, we did not change the text that describes our reasoning for using cycloheximide in our experiments in the Introduction and Results sections.

35 4. The subheading on Page 9 seems to apply to the previous figures. It would be best to add "in vitro" at the end to distinguish this section from the previous sections.

Excellent point. We made the requested change in the revised paper on page 10, line 5.

40 5. In Figure 5B, the accumulation of Hxt3-GFP at the boundary in the cell treated with CHX is not at all obvious. Also, the quantitative impact of CHX or HS on boundary localization in Figure 5C and on degradation in Figure 5E are not so obvious. This should be admitted in the text, and likely reflects the already robust sorting and cleavage in the in vitro assay induced by the

spheroplast procedure; i.e., this does not affect the main conclusions of the paper. Again, degradation of Hxt3-GFP in *Vps36delta* cells, particularly in the left-hand panel of Figure 5E, is not at all impressive and should be qualified in the text.

5 Excellent points! To address this concern, we replaced the micrograph in Figure 5B with another that better shows Hxt3-GFP accumulation in the boundary and added the following clarifications to the Results section of the revised paper:

Page 11, line 14: *“However, these responses were not as robust as those observed in vivo (see Figures 3 and 4). We reasoned that this is because Hxt3-GFP found on isolated vacuole membranes is already marked for degradation, as its presence is the product of down-regulation presumably triggered by glucose withdrawal during the organelle isolation procedure (Figure 5A) and it is degraded during fusion in vitro under control (unstimulated) conditions (Figure 5B – E). Thus, heat stress or CHX further enhances clearance of Hxt3-GFP that is already destined for degradation. Because these responses are additive, we reasoned that heat stress and CHX likely target distinct mechanisms, at least in part, from those that respond to glucose withdrawal. Because heat stress and CHX trigger Hxt3-GFP degradation in vitro, the underlying machinery must be present on vacuole membranes, whereas the machinery that senses glucose withdrawal must be, in part, present on the plasma membrane to accommodate endocytosis, and neither likely include ESCRTs.”*

The effects of the Ypt7 inhibitors on degradation appear to be quite variable (e.g. HS still increases degradation in the last lanes of the left bottom panel of Figure 5E and CHX still somewhat increases degradation in the last lanes of the right top panel). This should be qualified or better explained.

Agreed. To address this concern, we added the following explanation to the Results section of the revised paper:

Page 11, line 27: *“Next, we pretreated in vitro vacuole fusion reactions with Ypt7 inhibitors and found that they blocked Hxt3-GFP sorting, internalization and degradation (Figure 5B – E), implicating Ypt7 and its effectors in Hxt3-GFP sorting into boundary membranes. However, closer examination of western blots revealed that some residual Hxt3-GFP cleavage occurred in the presence of inhibitors under all conditions tested (Figure 5E; compare 0 to 120 minutes with fusion inhibitors, + F.I.). One explanation is that perhaps some vacuoles were docked, and contained Hxt3-GFP in their boundary membranes, upon isolation (i.e. they were engaged in the fusion process in live cells before lysis). If so, then Ypt7 inhibitors would be unable to block subsequent Ypt7-independent stages of fusion responsible for Hxt3-GFP internalization and degradation, accounting for the observed residual GFP-cleavage. This also accounts for the small fraction of the vacuole population that contains relatively high levels of Hxt3-GFP fluorescence in the boundary and lumen in the presence of Ypt7 inhibitors (Figures 5C and D). Thus, all things considered, we are confident that Ypt7 and the ILF pathway machinery located*

on the vacuole membrane is *likely* responsible for Hxt3-GFP degradation triggered by diverse stimuli.”

5 It is worth noting that we selected concentrations of Gyp1-46 and Gdi1 reported to completely block vacuole content mixing in vitro. However, results from in vitro homotypic vacuole fusion assays (both lipid mixing or content mixing) reflect only fusion events between vacuoles that make contact and fuse after isolation in vitro. This is not true for results shown in Figure 5, which also reflect tethering or docking events that may have occurred prior to organelle isolation.

10
6. The only additional experiment that should be done is an in vivo experiment with heat shock or cycloheximide treatment to complement the in vivo assay shown in Figure 6 to assess the efficiency of Hxt3-GFP clearance from the plasma membrane and sorting into the lumen of the vacuole.

15
This is an excellent experimental suggestion but we already present results from a similar time course experiment as the one requested in Figure 1 to show clearance of Hxt3-GFP from the plasma membrane and sorting into the ILF pathway in response to 2-deoxyglucose. We also
20 conducted a similar analysis as in Figure 6B and C, showing the progressive redistribution of Hxt3-GFP from vacuole membranes to the lumen over time (Figure 1E), which is an indicator of clearance efficiency. However, we cannot conduct the same analysis (as the one presented in Figure 6) to assess the efficiency of Hxt3-GFP clearance from vacuole membranes because the rate of homotypic vacuole fusion has not been accurately measured in vivo, at least to our
25 knowledge. The efficiency of Hxt3-GFP delivery to vacuole membranes by endocytosis would also affect this estimate further complicating the requested analysis.

30 Moreover, experiments shown in Figures 3 and 4 are very similar to the requested experiments whereby results show the clearance of Hxt3-GFP from the plasma membrane and sorting into the vacuole lumen in response to heat stress or cycloheximide in vivo, but only show a single time point. However, we did not provide an assessment of Hxt3-GFP clearance from the plasma membrane.

35 Thus, to address this concern, we conducted further analyses of micrographic data shown in Figures 3A and 4A whereby we calculated the distribution of Hxt3-GFP within the cell population before and after heat stress or cycloheximide, to provide an assessment of Hxt3-GFP clearance from plasma membranes in vivo. We also assessed its distribution on vacuole membranes and within the vacuole lumen of live cells to further support data shown in Figures 3C, 3D, 4C and 4D. These new results are shown in Figures 3B and Figure 4B of the revised
40 paper and support our conclusion that Hxt3-GFP is cleared from plasma membranes and sorted into the ILF pathway for degradation in response to heat stress or cycloheximide.

7. In Figure 7, again, while the sorting of Aqr1-GFP and Itr1-GFP into vacuole membranes, boundary membranes and the vacuole lumen in a *vps27*-independent way in panels A-D is obvious, the degradation upon heat shock in panel E is not. Itr1-GFP is barely degraded at all in wild-type cells, and HS-dependent degradation of Aqr1 is eliminated by *vps27* deletion. These data do not support the authors' conclusions regarding degradation. Note, the Figure legend refers at one point to *vps36delta* cells, whereas in this experiment *vps27delta* cells are used.

Good point. To address this concern, we repeated these experiments and replaced western blots in Figure 7E with new data that better show losses of full-length Aqr1-GFP or Itr1-GFP signal and increases in cleaved GFP signal after heat stress, supporting our original conclusion. We also corrected the figure legend to indicate that *vps27Δ* cells were studied.

8. Following points in the discussion need to be addressed:
- generally, the discussion is highly speculative and reads more like a commentary than a discussion of the contextual placement of the data in the paper within the field. I would recommend trimming the discussion substantially to focus on the contribution this paper makes rather than speculation on unaddressed issues, such as sorting into the ILF pathway, roles of lipids or ubiquitylation, etc. These are interesting points, but inappropriate for a discussion of this paper.

Good point. In our defense, this paper was reviewed twice before submission to *Nature Communications* and most discussion points were requested by previous reviewers. However, we concur, and to address this concern removed an entire section in the Discussion entitled "Possible basis of client protein selection" that addressed possible mechanisms of sorting including lipids and ubiquitylation.

- page 14, top: super-resolution microscopy allows distinction between intraluminal vesicles and the lysosomal limiting membrane, including STED which can be used for live cell microscopy.

Agreed. In theory, STED could image ILVs with diameters near 100 nm within MVBs of live cells but, to our knowledge, this has not been demonstrated and no references were provided. Thus, we limited our discussion to include methods exclusively used in previous published reports. With this in mind, we removed the following text from Discussion suggesting that CLEM may also be used to image ILV formation, as this has not been (convincingly) reported:

Page 15, line 3: "...samples must be fixed and imaged using electron microscopy coupled with immune-gold labeling ~~(or possibly correlative light electron microscopy).~~"

- page 14, bottom: re-evaluation of the GFP release data in this paper supports the contention raised here that disabling the ESCRT pathway can interfere with vacuolar/ lysosomal degradation.

We believe this is related to Concern #3 and directed at the following text in the Discussion that we changed to address both concerns:

- 5 Page 15, line 18: *“For example, in addition to blocking ILV formation, deleting ESCRT genes partially inhibits MVB-vacuole fusion (Karim et al., 2018), which is required to deliver internalized proteins to the ILF pathway, and prevents efficient delivery of some biosynthetic cargoes to the vacuole lumen (e.g. carboxypeptidase S and other vacuole hydrolases; Henne et al., 2011). This perhaps explains why Hxt3-GFP accumulates on puncta and the vacuole*
- 10 *membrane in vps36Δ cells (e.g. Figure 1C – E) and its degradation profile is different than in wild type cells (Figures 1F, 3F, 4F, 5E and 7E).”*

REVIEWERS' COMMENTS:

Reviewer #1 (Remarks to the Author):

The authors addressed all the points I had raised in the previous review process. The immunoblot data now hold consistence, and I am convinced that the uniformity of the membrane protein localizations on the surface of a single vacuole is not necessarily a prerequisite for the sorting by the ILF pathway.